# A sexually dimorphic pre-stressed translational signature in CA3 pyramidal neurons of BDNF Val66Met mice

Jordan Marrocco [1], Gordon H. Petty[1], Mariel B. Ríos[1], Jason D. Gray[1], Joshua F. Kogan[1], Elizabeth M. Waters[1], Eric F. Schmidt[2], Francis S. Lee[3] & Bruce S. McEwen[1]

Males and females use distinct brain circuits to cope with similar challenges. Using RNA sequencing of ribosome-bound mRNA from hippocampal CA3 neurons, we found remarkable sex differences and discovered that female mice displayed greater gene expression activation after acute stress than males. Stress-sensitive BDNF Val66Met mice of both sexes show a pre-stressed translational phenotype in which the same genes that are activated without applied stress are also induced in wild-type mice by an acute stressor. Behaviourally, only heterozygous BDNF Val66Met females exhibit spatial memory impairment, regardless of acute stress. Interestingly, this effect is not observed in ovariectomized heterozygous BDNF Val66Met females, suggesting that circulating ovarian hormones induce cognitive impairment in Met carriers. Cognitive deficits are not observed in males of either genotype. Thus, in a brain region not normally associated with sex differences, this work sheds light on ways that genes, environment and sex interact to affect the transcriptome's response to a stressor.

[1] Laboratory of Neuroendocrinology, The Rockefeller University, 1230 York Avenue, New York, NY 10065, USA. [2] Laboratory of Molecular Biology, The Rockefeller University, 1230 York Avenue, New York, NY 10065, USA. [3] Sackler Institute for Developmental Psychobiology, Department of Psychiatry, Weill Cornell Medical College, 1300 York Avenue, New York, NY 10065, USA. Correspondence and requests for materials should be addressed to B.S.M. (email: mcewen@rockefeller.edu)

Sex differences in the brain are not limited to the control of reproduction, but involve a variety of physiological, neuroanatomical and behavioural functions that develop through hormonal and genetic factors, including the role of mitochondria from the mother[1]. Because females are generally underrepresented in basic and preclinical studies[2], the scientific community has advocated that sex-based differences should be incorporated as a biological variable in neuroscience research[3–5]. This is particularly relevant to the study of brain and behaviour, as males and females use distinct brain circuits to cope with similar challenges[6–8]. The effects of environmental stress may induce an increased likelihood of developing psychiatric disorders that differ in frequency between males and females[5, 9].

The hippocampus, a brain region that regulates memory and emotions, is a crucial target of stress and allostatic load, the cumulative burden of experiences and health behaviours[10, 11], both because of its neuroanatomical connections and because of its expression of glucocorticoid and mineralocorticoid receptors[1, 12–14]. While not widely regarded as a target of sex hormone actions, the hippocampus expresses oestrogen, androgen and oestrogen-inducible progestin receptors, and responds to stressors in ways that differ between males and females[1, 15, 16]. The CA3 is a region of the hippocampus that has a crucial role in the stress response as it exerts a negative feedback in the regulation of the hypothalamic-pituitary-adrenal (HPA) axis[17]. The CA3 region is also a nexus of information processing of spatial and episodic memory[18] and a site of dynamic stress-induced structural plasticity[1, 19]. To give a snapshot of how a specific neuron type within a stress-responsive and vulnerable brain region responds, we have focused on the effects of an acute stress challenge in neurons of this region.

Recently, the discovery of a common human single nucleotide polymorphism (SNP) in the brain-derived neurotrophic factor (BDNF) gene, which results in an amino acid change from a valine to a methionine at position 66 (Val66Met), has been implicated as a modifier of neuropsychiatric disorders[20, 21]. Heterozygous carriers of the variant BDNF SNP have a frequency between 4% and 40%, depending on ethnicity[22–24]. A knock-in mouse with the variant BDNF Met allele recapitulates many of the phenotypic hallmarks of $BDNF_{Met}$ carriers with regards to both brain anatomy, as well as behaviour. Neurons expressing the BDNF Met allele show reduced activity-dependent BDNF secretion when compared with neurons expressing BDNF Val allele[21, 25]. Mice carrying the Met allele display exacerbated stress response, cognitive impairment and greater anxious-depressive behaviours[26], which in $BDNF^{Met/Met}$ females critically interact with the oestrus cycle[27, 28]. This suggests that the BDNF Met allele interacts with gonadal hormones to control brain function and that biological sex impacts the stress response in the hippocampus.

Recently, the translational profiles of $BDNF^{+/+}$ and $BDNF^{Met/+}$ male mice subjected to chronic stress revealed highly distinct molecular changes within CA3 pyramidal neurons by genotype[29]. This study used a transgenic mouse line to isolate in vivo translating messenger RNA (mRNA) from a genetically homogenous population of CA3 pyramidal neurons. Of note, BDNF is enriched in the mossy fibre pathway that projects to CA3 neurons[30], which themselves also display sex differences in dendritic remodelling in response to stress[31]. Remarkably, remodelling of CA3 pyramidal neurons is also observed after chronic glucocorticoid treatment, indicating that activation of the HPA axis is involved in causing retraction of CA3 neurons[32].

Using RNA sequencing (RNA-seq) data from a single neuronal cell-type, pyramidal CA3 neurons, we explored sex, as well as BDNF genotypic differences on transcriptional regulation in response to acute stress. We report not only that the BDNF Val66Met variant confers differential sensitivity to acute stress that differs between males and females, but also that the sex and genotype differences are much larger than expected in the CA3 neurons.

## Results

**Sex-dimorphic gene expression in response to acute stress.** To assess the sex-specific effects of acute stress on the translational profile of CA3 pyramidal neurons of the hippocampus, we used the translating ribosome affinity purification (TRAP) method to isolate cell type-specific translating mRNAs[33]. BAC transgenic mice expressing EGFP-tagged ribosome protein L10a (EGFP-L10a) specifically in CA3 pyramidal cells were used. EGFP-tagged polysomes were immunoprecipitated (IP) from hippocampal homogenates and bound mRNA was isolated and analysed by high throughput RNA sequencing. The TRAP immunoprecipitation and the poly A library prep are two independent steps designed to obtain a homogeneous transcriptome from CA3 neurons, and eliminate rRNA and any small RNAs. This substantially decreases the diversity of the transcriptome and thereby improves power at equivalent sequencing depths[33].

We found that acute stress protocol activated two immediate-early genes, Fos and Arc, in both males and females (Fig. 1a). Yet there was a considerable sex dimorphism in the expression of other genes, as RNA-sequencing in proestrus females, when circulating oestradiol levels are higher, revealed 6472 genes altered by acute stress in CA3 neurons compared with unstressed controls. In contrast, males exhibited a markedly lower number of genes (2474) affected by acute stress in CA3. A comparison of genes altered by acute stress in both males and females found that 1842 genes were affected by stress in both sexes. However, separating the stress-regulated genes in females and males, based on the direction of their fold change, revealed that the number of stress-regulated genes that are common to both sexes dramatically decreased to only 21 genes in the up-regulated list and 44 genes in the down-regulated list (Fig. 1b, Supplementary Fig. 1). Overlapping genes included the immediate early genes, Arc and Fos, as well as several genes previously implicated in the acute stress response (Supplementary Fig. 1).

This indicates that the vast majority of overlapping genes are up-regulated by stress in males and down-regulated in females, or vice-versa. Indeed, we found 649 genes that were up-regulated by stress in males and down-regulated in stressed females, and a further 1139 genes that were up-regulated by stress in females and down-regulated in stressed males. A gene ontology analysis revealed that those genes were implicated in synaptic function, transcription regulation, glycosylation and ion channel (Table 1). The heat map cladogram of the 100 most differentially regulated genes among all groups highlighted a greater stress-induced transcriptional change in females than in males, with stressed females clustering furthest from control females, stressed males, and control males (Fig. 1c).

This analysis was also performed on ribosome-bound transcripts isolated from CA3 pyramidal neurons of females in metoestrus/dioestrus, when circulating oestradiol levels are lower. Comparing the effects of stress in high vs. low oestradiol females, sex differences largely surpassed differences between oestrus cycle stages. High and low oestradiol females displayed 4676 genes that were commonly altered by acute stress, comprising 86.8% of genes altered in high-oestradiol females and 75.3% of genes altered in low oestradiol females (Supplementary Fig. 2A). A heat map of the 100 most differentially regulated genes confirmed that females displayed the same expression profile in response to acute stress regardless of the oestrus cycle stage (Supplementary Fig. 2B). Thus, we focused our analysis on females in proestrus.

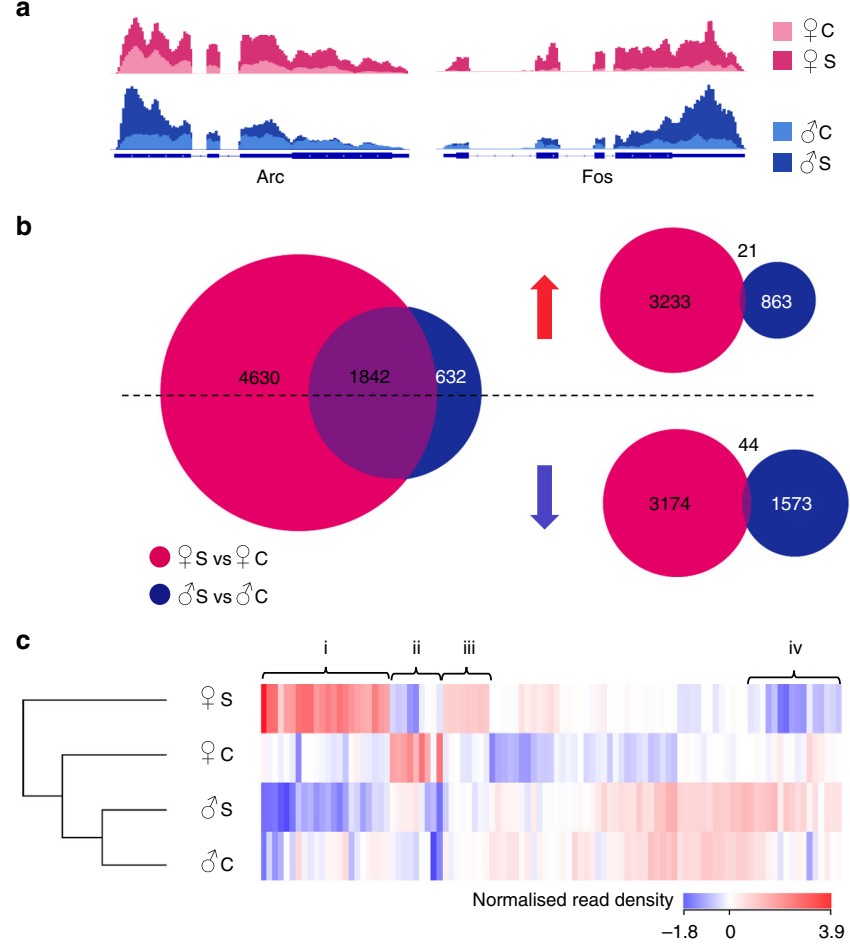

**Fig. 1** Acute stress affects CA3 neurons in a sex-dependent manner and alters a greater number of genes in females than in males. **a** Histogram of mapped reads showing expression of *Arc* and *Fos*, as genes representative of the immediate early gene transcription (IEG) cascade. Blue line represents the introns (thin) and exons (thick). Shaded areas indicate the number of normalized reads for female controls (light pink), stressed female (dark pink), male controls (light blue), and stressed male (dark blue). As expected, acute stress increases IEG expression in both males and females. **b** (Left) Venn diagram depicting the number of genes altered by acute stress in females (pink circle), males (blue circle), and in both sexes (purple overlap) (Z-score < 0.001; absolute fold change > 1.5). (Right) Venn diagrams are broken into up-regulated (red arrow) and down-regulated (blue arrow) genes. **c** Heat map representing the normalized read density of the 100 genes with the highest variance across all groups. Genes were clustered based on similar expression profile: (i) Genes up-regulated by stress in females and down-regulated by stress in males; (ii) Genes down-regulated by stress in females and unaffected by stress in males; (iii) Genes up-regulated by stress in females and unaltered by stress in males; (iv) Genes down-regulated by stress in females and up-regulated by stress in males. C control, S stressed

A gene set enrichment analysis (GSEA) was used to cluster stress-related genes, i.e., genes induced by stress, based on their biological function. The normalized read density of each gene was used to compute an enrichment score for each cluster. The top 10% of the most enriched pathways were then organized in a network using the Cytoscape® software (see Methods section). This placed clusters with several shared genes together, allowing for visualisation of the overall regulation of biological functions in the CA3 neurons of stressed male and female mice. There were fewer stress-enriched gene sets in males than in females. Stress-biased genes in males largely belonged to transcriptional regulation and synaptic function. Using Database for Annotation, Visualization and Integrated Discovery (DAVID) functional annotation cluster tool, we clustered genes based on gene ontology. In females stress-biased genes belonged to a wider array of biological functions, such as neurogenesis, cytoskeleton, excitatory neurotransmission, morphogenesis, and intracellular trafficking, in addition to those found in males (Fig. 2a, b). Again, the pathways that were found to be up-regulated by stress in males were largely down-regulated by stress in females (Fig. 2b).

**Pre-stressed translational profiling of BDNF Met carriers**. To examine sex differences in the translational profile of CA3 neurons in a model that recapitulates genetic susceptibility to stress-induced neuropsychiatric disorders, Gprin3 BAC-TRAP mice were crossed with BDNF Val66Met mice. This knock-in mouse model carries a single-nucleotide polymorphism that is a modifier of psychiatric disorder vulnerability in humans[20, 34]. Using the TRAP method, we isolated CA3 ribosome-bound mRNA transcripts from male and female mice that were either heterozygous for the Met allele (BDNF$^{Met/+}$) or did not posses the Met variant (BDNF$^{+/+}$). A subset of BDNF$^{Met/+}$ and BDNF$^{+/+}$ mice had been exposed to an acute stress paradigm to investigate the interaction between stress and the Met variant.

In both genotypes and sexes, acute stress activated the expression of cFos in CA3 neurons, as shown by the read density mapped to the *Fos* gene in the TRAP-seq data (Fig. 3a), demonstrating the efficacy of the acute stress protocol. Comparing genotypes, TRAP-seq revealed that, in BDNF$^{Met/+}$ females, 6,882 genes were altered, while in BDNF$^{Met/+}$ males only 1,415 genes were altered, when compared with same-sex BDNF$^{+/+}$

**Table 1 Genes differentially regulated by stress in female mice versus males**

| GO term | Enrichment score | Total genes | Select genes |
|---|---|---|---|
| *Upregulated in females and downregulated in males* | | | |
| Cytoplasmic membrane | 67.72 | 649 | *L1cam, Grin3a, Grin2a, Scn1a, Kcnj3, Kcnj4, Kcnj6, Mmp17, Npy2r, Oprl1, Htr2a, Cx3cl1, Astn2, Grin2, Shank2, Lamp2, Gabbr1, Gabbr2, Creb3, Rapgef2, Fgfr1, Gabra1* |
| Glycosylation/glycoprotein | 36.98 | 564 | *L1cam, Syp, App, Grin2a, Oprl1, Syt11, Syt12, Slc7a1, Gabbr1, Gabbr2, Slc1a4, Lc1a6, Dag1, Icam5, Grm1, Neto1, Sort1, Grin1, Lamp1, Lamp2, Cacna1a, Cacna1b, Gabbra1, Astn1, Slit1, Slit2* |
| Endoplasmic reticulum | 15.68 | 125 | *Sort1, Nr3c2* |
| Ion channel/ion transport | 10.99 | 78 | *Grin3a, Grin1, Grm7, Grin2a, Kcnq, Kcnq3, Kcnq5, Kcng1, Cacna1a, Cacna1b, Cacna1c, Cacna1e* |
| Lysosome | 9.74 | 48 | *Lamp1, Lamp2, Sort1, Clcn7* |
| Cell–cell adhesion | 8.65 | 82 | *L1cam, Celsr3, Amigo2, Nell1, Nell2, Cntn1, Cntn2, Cntn3, Astn1, Nlgn1, Nlgn3* |
| Synaptic vesicle | 8.41 | 106 | *Gabbr1, L1cam, Gabbr2, Grin3a, Rims1, Grin1, Grin2a, Shank2, Grm1, Gria2, Grm7, Sort1, Nlgn1, Shank2, Grin2a, Sema4f* |
| Synapse | 8.00 | 85 | *Syt1, Gabbr1, Gabbr2, Grin3a, Nrcam, Svt2a, Grin2a, Grm1, Grm7, Grin1, Nlgm1, Nlgm3, Shank2, Hdac4, Gria2, Syt1, L1cam, Rims1, Rims3, Rims4, Gabbrg2* |
| *Downregulated in females and upregulated in males* | | | |
| Ribosome | 15.9 | 116 | *Mobp, Brca1, Alf1, Tgfb1l1, Adora2a, Setd8* |
| Transcription regulation | 3.86 | 150 | *Sox11, Pou3f2, Brca1, Tgf B1, Bcl2, Tnfsf12, Fgf1, Igf1, Adora2a, Tgfb1l1, Stk11, Nr2e1* |
| Telencephalon development | 3.43 | 17 | *Cx3cr1, Pou3f2, Nr2e1, Foxp2, Fabp7* |
| Cytoskeleton | 3.14 | 22 | *Tppp3, Prca1, Bcl2, Shc1, Gadd45a* |

Genes that are up-regulated by stress in females and down-regulated in males, and vice-versa, were obtained by performing a fold-change analysis using Strand NGS. Columns represent (i) Gene ontology term, (ii) enrichment score and (iii) total number of genes obtained from DAVID annotation clustering; (iv) genes selected based on their known role in neuronal function. Several of the most enriched GO terms were selected based on their implication in neuronal and epigenetic function

mice. Again, more genes were up- or down-regulated by stress in females than in males. Of these genes 1044 were commonly altered by the Met allele in both sexes. Within the BDNF^Met/+ genotype, we observed that only 40 of these genes were up-regulated in both males and females, and only 31 were down-regulated in both sexes (Fig. 3b).

A gene ontology (GO) analysis was performed on the genes that were up-regulated in female BDNF^Met/+ mice and down-regulated in male BDNF^Met/+ mice, and vice-versa. This analysis revealed that these genes were implicated in biological processes such as synaptic function, transcriptional regulation, glycosylation, neuronal architecture, and ion channels (Table 2).

We next examined gene expression changes across genotypes. The regulation of gene expression in BDNF^Met/+ mice mimicked the sex-dimorphic translational response to acute stress in CA3 neurons of wild-type mice (Fig. 1b). This led us to hypothesize that the stress-induced genes may overlap with genes expressed without applied acute stress in BDNF^Met/+ mice, indicating a pre-stress state in this genotype that does not involve immediate early gene activation.

To characterize the pre-stress state, we compared the list of genes affected in acutely stressed BDNF^+/+ mice and in unstressed BDNF^Met/+ mice to determine which genes were altered in both groups. In females, out of the 6484 stress-sensitive genes in acutely stressed BDNF^+/+ mice and the 6882 Met-allele-altered genes in unstressed BDNF^Met/+ mice, 73% of genes (5621) were common to both groups. In males, stressed BDNF^+/+ mice and BDNF^Met/+ mice displayed an overlap of 38% of genes (Fig. 3c). The heat map cladogram of the 100 most differentially

regulated genes among all groups emphasized a large similarity in the CA3 transcriptional profile between stressed BDNF^+/+ mice and unstressed BDNF^Met/+ mice, with the two groups clustering together in both sexes. However, the similarity between stress-induced genes and Met-regulated genes in CA3 neurons was greater in females than in males (Fig. 3d).

The pre-stress state genes differed considerably between males and females. Using GSEA, stress-biased genes and Met-regulated genes were clustered based on their biological function. The read density of each gene was used to calculate an enrichment score for each biological cluster, for both stressed BDNF^+/+ and unstressed BDNF^Met/+. The top 10% most enriched pathways revealed that stress and the Met allele altered more pathways in female mice than in male mice (Fig. 4a). Furthermore, in males, genes common to both stressed BDNF^+/+ and unstressed BDNF^Met/+ largely belong to epigenetic regulation and small ribosomal subunits, and to a lesser extent, transcription factor regulation (Fig. 4a). In females there was a larger overlap in biological functions than in males that is shared between stress-biased genes and BDNF_Met-regulated genes. These genes were involved in biological processes such as protein glycosylation, cellular homoeostasis, cytoplasmic secretory vesicles, axon projection guidance, learning and memory, synaptic structure, neuronal cell body, and transmembrane ion transport (Fig. 4a).

A selection of the eight most enriched GO clusters from both male and female mice revealed that clusters that are up-regulated in females by either stress or pre-stress in the presence of the BDNF_Met allele are down-regulated in males, and vice versa

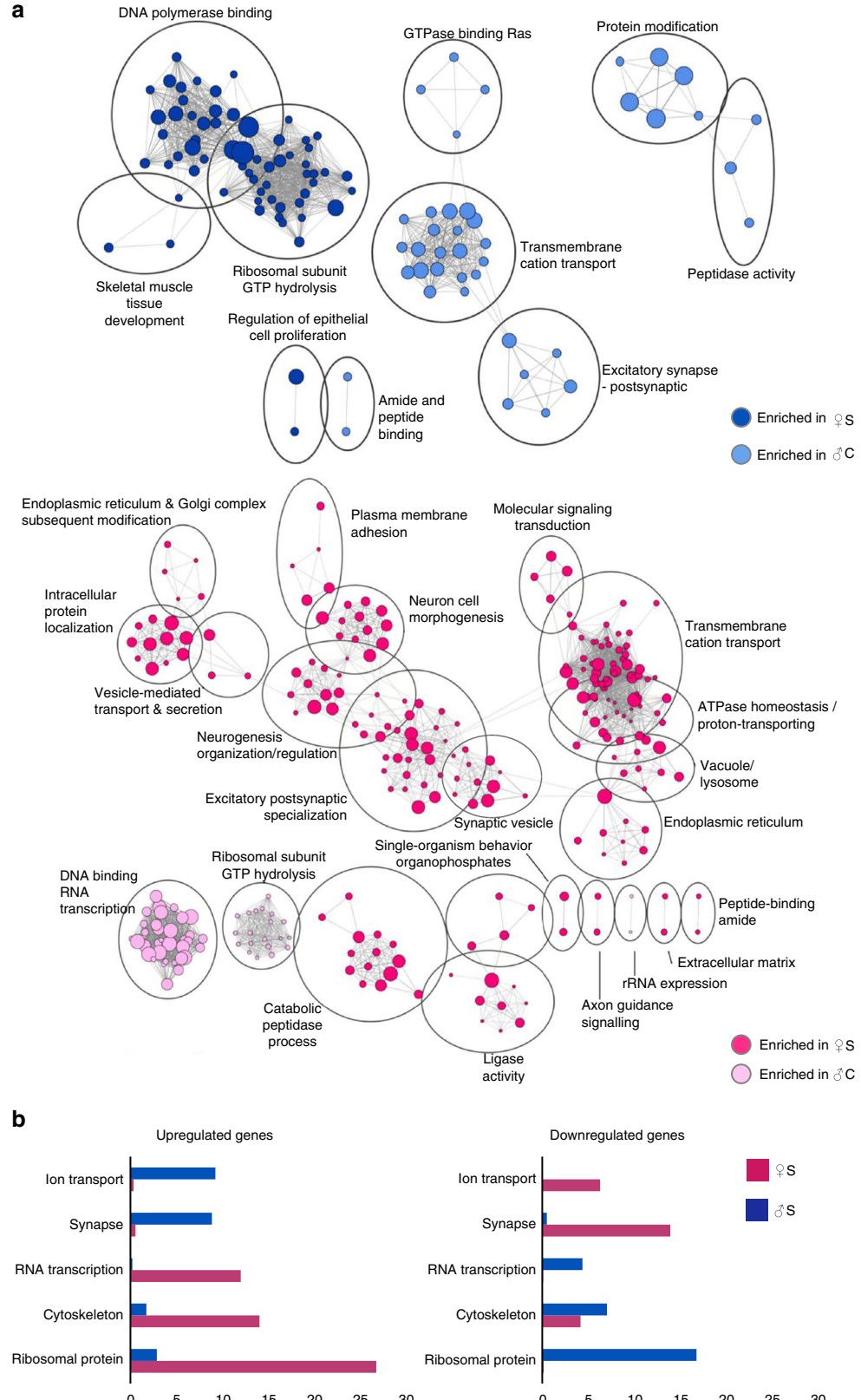

**Fig. 2** Acute stress regulates sex-dimorphic pathways in CA3 neurons and induces more pathways in females than in males. **a** Gene set enrichment map of the top 10% most enriched pathways in stressed males or females, i.e., up-regulated by stress (dark blue circles and dark pink circles, respectively) or in control males or females, i.e., down-regulated by stress (light blue circles and light pink circles, respectively) (Z-score < 0.001; absolute fold change > 1.5). Circle size correlates to the number of genes in the pathway. Grey lines connect pathways that share the same genes. Pathways are clustered based on number of shared genes and named based on biological function. **b** Database for Annotation, Visualization and Integrated Discovery (DAVID) enrichment scores. Up-regulated (left) and down-regulated (right) genes in stressed females (pink) and stressed males (blue) are clustered based on the GO scores and given an enrichment score. Five GO terms showing the highest enrichment score are presented as a bar graph. C control, S stressed

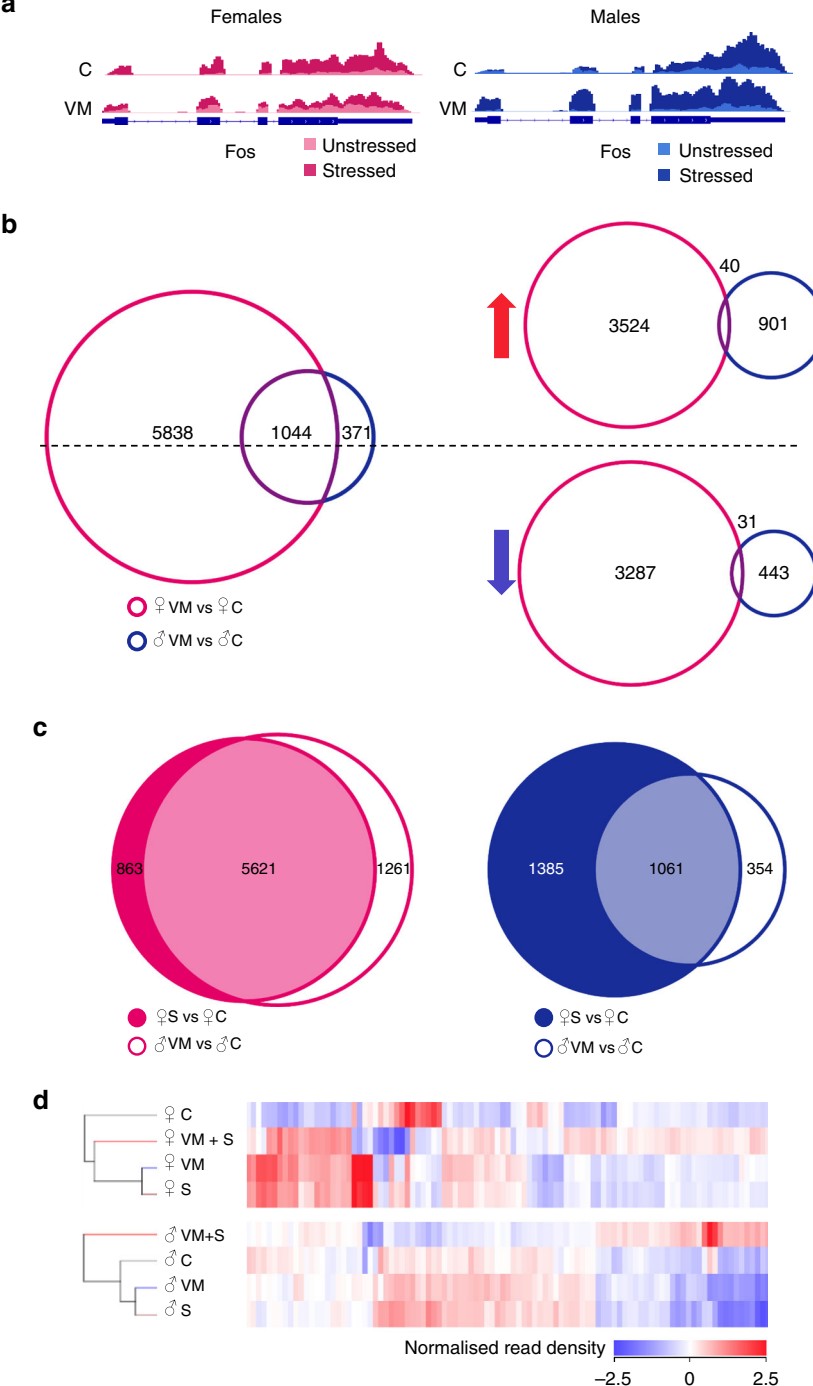

**Fig. 3** Male and female BDNF[Met/+] mice show a similar transcriptional profile to stressed BDNF[+/+] mice in CA3 neurons. **a** Histogram of mapped reads for the gene encoding c-Fos (*Fos*). Blue lines represent the introns (thin) and exons (thick). Shaded areas indicate the number of normalized reads for female controls (light pink), stressed females (dark pink), male controls (light blue) and stressed males (dark blue), for both BDNF[+/+] and BDNF[Met/+] mice. Acute stress is required to elevate c-Fox expression. **b** (Left) Venn diagram depicting the number of genes altered in unstressed BDNF[Met/+] females (pink circle), males (blue circle) and in both sexes (purple overlap) (Z-score < 0.001; absolute fold change > 1.5) when compared with unstressed BDNF[+/+]. (Right) Venn diagrams are broken into up-regulated (red arrow) and down-regulated (blue arrow) genes. **c** (Left) Venn diagram depicting the number of genes affected in stressed BDNF[+/+] (dark pink circle) or unstressed BDNF[Met/+] females (white circle), or altered in both groups (light pink overlap) when compared with unstressed BDNF[+/+] females (Z-score < 0.001; absolute fold change > 1.5). (Right) Venn diagram representing the number of genes affected in stressed BDNF[+/+] males (dark blue circle) or unstressed BDNF[Met/+] males (white circle), or altered in both groups (light blue overlap) when compared with control males (Z-score < 0.001; absolute fold change > 1.5). **d** Heat map representing the normalized read density of the same 100 genes, as in Fig. 1c across all groups in both males and females. The cladogram to the left indicates the similarity in gene expression profiles for each group. C unstressed BDNF[+/+], S stressed BDNF[+/+], VM unstressed BDNF[Met/+], VM + S stressed BDNF[Met/+]

(Fig. 4b). This indicates that both acute stress in BDNF[+/+] mice and the pre-stress state in the BDNF[Met/+] mice induced distinctly different patterns of gene expression in males and females in CA3 neurons.

**Gene expression patterns**. To assess the distinct expression patterns of males and females, genes were thresholded based on the magnitude and direction of their fold change in stressed BDNF[+/+], unstressed BDNF[Met/+], and stressed BDNF[Met/+] mice. Genes with the same expression profile were clustered together. Out of the 27 possible expression profiles (see Methods), we highlighted four in which genes were similarly regulated in stressed BDNF[+/+] and unstressed BDNF[Met/+], i.e., genes were up-regulated or down-regulated in both groups. In females, these groups accounted for 1675 genes (17.6%), and in males 1004 (16.3%). To investigate transcripts included in these profile clusters, we performed a GO analysis on the genes found in each cluster. In one pattern of expression, genes were down-regulated in both stressed BDNF[+/+] and unstressed BDNF[Met/+], but unaltered in stressed BDNF[Met/+] (Fig. 5a). This cluster showed enrichment in chromatin structure, cytoskeleton, and cellular response to stress in females, whereas in males, ion binding and exocytotic/transport machinery were enriched. Transcriptional regulation was enriched in both sexes.

Another cluster, in which genes were up-regulated in stressed BDNF[+/+] and unstressed BDNF[Met/+] and unaltered in stressed BDNF[Met/+], also showed enrichment of cytoskeleton GO terms in females, as well as males (Fig. 5b). In addition, cell/cell adhesion as well as protein transport/localisation GO terms were enriched in females, and transcription and RNA processing GO terms were enriched in males. A third profile had genes up-regulated in stressed BDNF[+/+] and unstressed BDNF[Met/+], but down-regulated in stressed BDNF[Met/+] (Fig. 5c). This profile again revealed enrichment of cytoskeleton GO terms, but only in females. Females also showed enrichment of cell–cell adhesion/extracellular matrix, as well as calcium binding. In males, ion homoeostasis, as well as transcription, protein, and chromatin regulation terms were enriched. Interestingly, one expression profile existed only in males, namely, one in which genes were down-regulated in stressed BDNF[+/+] and unstressed BDNF[Met/+], but up-regulated in stressed BDNF[Met/+] mice (Fig. 5d). This profile displayed particularly high enrichment for membrane and glycoprotein/glycosylation terms. As previously noted, the cluster in which genes were selectively up-regulated in stressed BDNF[+/+] and stressed BDNF[Met/+] includes genes belonging to the immediate early transcription cascade such as *Fos* and *Arc* (Fig. 5e).

Remarkably, in females, a large portion of the pre-stress genes were up-regulated in both stressed BDNF[+/+] mice and BDNF[Met/+] mice (Fig. 5b), whereas, in males, a majority of pre-stress genes were down-regulated in an expression pattern that was not found in females (Fig. 5d). This indicates that, despite the similarities between BDNF[Met/+] mice and stressed BDNF[+/+] mice, the BDNF Met allele does not induce the immediate early gene transcriptional cascade and that an acute stressor is required.

**Sex-dimorphic regulation of the glutamate/GABA system**. Many of the common genes identified by GO and GSEA analysis in stressed wild-type mice and BDNF Met allele carriers are implicated in glutamatergic and GABAergic neurotransmission (Table 2). For example, the NMDA receptor subunit genes *Grin2a* and *Grin3a* were both up-regulated in stressed wild-type and unstressed BDNF[Met/+] female mice. Similarly, the metabotropic glutamate receptors *Grm1* and *Grm7* were up-regulated by stress in females, and *Grm5* was up-regulated in unstressed

BDNF[Met/+] female mice. Interestingly, the *Gabra1* gene, which encodes a GABA receptor subunit, showed the same pattern. In addition to neurotransmitter receptors, genes implicated in neurotransmitter release, such as *Rims1* and the Ca + channels *Cacna1b* and *Cacna1c*, were altered in both of these groups. Interestingly, males showed the opposite response profile of the NMDA and GABA genes, and up-regulation of the genes implicated in transcription and translation, such as the transcription factors *Sox11* and *Dlx1* and the DNA binding proteins *Hmg1* and *Hmg2*.

Beyond the neurotransmitter system, genes implicated in synaptic plasticity were changed in both groups. For example, the transmembrane cell adhesion molecule *L1cam* and the epigenetic modifying enzyme *Hdac4* both were increased in stressed BDNF[+/+] females and unstressed BDNF[Met/+] females. Both these changes, and those associated with neurotransmission, were unique to females and not observed in the intersection of these sets in male mice. Again, males show the opposite response and down-regulation of these genes. (Tables 1, 2).

**The pre-stress state and the regulation of GR-binding genes**. The hippocampus is known to be particularly sensitive to stress-induced glucocorticoid receptor (GR) activation, resulting in neuronal functional modulation[35, 36]. Recently, ChIP-sequencing identified a repertoire of GR-binding genes within the hippocampus that respond to the stress hormone corticosterone[36]. Because CA3 neurons of BDNF[Met/+] mice showed a similar transcriptional profile to those of acutely stressed BDNF[+/+] mice, we investigated whether the Met variant affected the GR-binding repertoire within CA3 neurons. Indeed, clustering male and female mice based on the expression of 1,550 GR binding genes showed that the CA3 transcriptional profile of unstressed BDNF[Met/+] males and females mimicked gene expression changes in stressed BDNF[+/+] mice (Fig. 6a). This clustering recapitulated that of Fig. 3d, where stressed BDNF[+/+] and unstressed BDNF[Met/+] clustered further from unstressed BDNF[+/+] mice in females than in males. Interestingly, in male mice, this clustering was observed despite the fact that the GR gene itself was not altered by stress or by the Met allele.

Levels of GR mRNA, as well as mRNAs for essential chaperone proteins were directly compared between stressed mice and BDNF[Met/+] mice. A significant reduction in GR mRNA was observed in stressed females, but not in male mice. Further, BDNF[Met/+] female showed a significant decrease in GR mRNA levels at baseline, whereas males did not. Mineralocorticoid receptor (MR) levels in females show the opposite pattern to that of GR, and also differed from that in males. Finally, the GR chaperones *Bag1* and *Fkbp5*, which can regulate GR translocation to the nucleus, were significantly decreased in stressed BDNF[+/+] females and in female Met allele carriers, but not changed in BDNF[+/+] male mice with stress or with the Met allele (Fig. 6b). Thus stress and the BDNF Met allele induced a greater change of GR binding gene expression in females than in males.

**Sex-specific spatial memory performance of BDNF Met carriers**. Stress-sensitive behavioural tests of spatial memory were conducted to address the in vivo outcomes of these molecular differences. In males, stress affects hippocampal CA3 neurons by inducing dendritic remodelling, ultimately leading to behavioural effects such as memory impairment[10, 12, 37, 38], whereas this does not occur in the same way in females[39]. Indeed, chronic restraint stressed female rats tested in the radial arm maze and object location tasks displayed enhanced spatial memory performance, whereas males were impaired in all memory tests[39]. This was associated with stress-increased levels of 5-HT and

**Table 2 Genes differentially regulated by the BDNF Met allele in female mice versus males**

| GO term | Enrichment score | Total genes | Select genes |
|---|---|---|---|
| *Upregulated by BDNF$^{Met/+}$ in females and downregulated in males* | | | |
| Cytoplasmic membrane | 23.15 | 377 | *L1cam, Grin3a, Grin2a, Clcn3, Rims1, Kcnj3, Sam5b, Celsr2, Clcn6, Clcn7, Cacng8, Grm5, Sema4f, Npy2r, Nlgn3, Astn1, Astn2, Cacnad2d1, Gabra1, Kcnb2, Shank2, Lamp1, Lamp2, Cacna1e, Cacna1c, Cacna1b, Kcnq3, Slit1, Slit2, Slit3, Dkk3* |
| Glycoprotein/glycosylation | 14.19 | 258 | *Ncam2, L1cam, Ephb6, Grin2a, Clcn3, Rims1, Sam5b, Celsr2, Clcn6, Cacng8, Grm5, Sema4f, Astn1, Astn2, Cacnad2d1, Gabra1, Kcnb2, Lamp1, Lamp2, Cacna1e, Cacna1c, Cacna1b, Slit1, Slit2, Slit3, Sema6b* |
| Lysosome | 6.88 | 113 | *Lamp1, Lamp2, Nos1* |
| Ion channel activity | 6.73 | 96 | *Grin3a, Kcnq3, Grin2a, Clcn3, Rims1, Kcnj3, Rims4, Kcns2, Gabra1, Kcnb2, Kcnj6* |
| Neuronal projection | 3.88 | 76 | *Nos1, Grin2a, L1cam, Gabbr2, Ncam2, Cacna1c, Cacna1b,* |
| Synapse | 3.27 | 160 | *L1cam, Gabbra2, Grin3a, Rims1, Rims4, Gabbra1, Nos1, Grin2a, Nlgn3, Hdac4, Gabra1, Sema4f* |
| *Downregulated by BDNF$^{Met/+}$ in females and upregulated in males* | | | |
| Transcription regulation | 4.56 | 81 | *Sox11, Hmg1, Hmg2, Lyn, Bcl2* |
| Ribosome | 2.98 | 22 | *Tpm2, Nufip2, Ctse, Rpl37, Rpl8, Rpl16* |
| Telencephalon development | 2.19 | 8 | *Cx3cr1, Dlx1, Fabp7* |

Genes that are up-regulated by the BDNF$^{Met/+}$ allele in females and down-regulated in males, and vice-versa, were obtained by performing a fold-change analysis using Strand NGS. Columns represent (i) Gene ontology term, (ii) enrichment score and (iii) total number of genes obtained from DAVID annotation clustering; (iv) genes selected based on their known role in neuronal function. Several of the most enriched GO terms were selected based on their implication in neuronal and epigenetic function

norepinephrine in the CA3 of females, but not males, and increased GABA in males, but not females[40]. The Met allele also interacts with the encoding of object location in the hippocampus, both in humans and experimental animal models[27, 41, 42].

Because transcriptional regulation of BDNF$^{Met/+}$ mice in CA3 neurons closely mimics that of acutely stressed BDNF$^{+/+}$ mice, we assessed whether the effect of the Met allele on cognitive behaviour recapitulated that of acute stress. The object placement paradigm was used to assess performance memory tasks (adapted from Ennaceur et al.[43, 44]). During the habituation phase in the open field, mice did not display any genotype effects in either the distance travelled or the time spent in the centre of the arena (Supplementary Fig. 3). Male and female mice showed no genotype effect in time spent exploring objects during the acquisition phase. During the recall day both sexes showed an increased time in exploring the objects after acute stress (females: $F_{(1,44)} = 10.36$, $P < 0.05$; males: $F_{(1,44)} = 15.07$, $P < 0.001$) (Fig. 7a). However, regardless of the stress-induced increase in exploration time, male mice displayed a lower discrimination index after acute stress regardless of genotype ($F_{(1,44)} = 4.41$, $P < 0.05$). This effect was not observed in female mice. Instead, BDNF$^{Met/+}$ female mice showed poor performance in discriminating the misplaced object, regardless of stress ($F_{(1,44)} = 8.15$, $P < 0.01$) (Fig. 7b). Finally, cognitive behaviour of unstressed BDNF$^{Met/+}$ mice did not recapitulate that of stressed BDNF$^{+/+}$ mice, regardless of sex. Thus behaviour of the animal does not reflect what we observe in CA3 neurons, implying that other brain regions are likely compensating and complementing each other.

Given that only BDNF$^{Met/+}$ females displayed cognitive impairment regardless of stress, we investigated whether the Met allele interacted with estradiol to affect memory. BDNF$^{Met/+}$ females and their matched BDNF$^{+/+}$ were ovariectomized and replaced either estradiol (300 nM in 0.1% ethanol) or vehicle (0.1% ethanol) in drinking water for 4 weeks. Ovariectomized BDNF$^{Met/+}$ females treated with vehicle did not exhibit memory impairment in the novel object placement test, resembling unstressed BDNF$^{Met/+}$ males. There was a trend for estradiol replacement to improve memory only in BDNF$^{+/+}$ but not in BDNF$^{Met/+}$, which still displayed no impairment in cognitive performance (Fig. 7b). These findings indicate that memory impairment in Met carriers is associated with endogenous circulating ovarian hormones. However, exogenous estradiol did not reestablish the impaired phenotype, suggesting a role for other ovarian hormones. Thus, the BDNF Met allele induced unique behavioural phenotypes in gonadal-intact males and females, with females exhibiting impaired memory at baseline.

## Discussion

While CA3 pyramidal neurons of the hippocampus are a major nexus of stress-induced changes in neuronal structure, memory function, and BDNF expression, it is not normally regarded as a target of sex differences or sex hormones. Yet the findings of the present study demonstrate that acute stress induces a remarkable array of sex- and genotype- specific translational profiles of mRNA isolated from CA3 pyramidal neurons, which are targets of BDNF[45]. The remarkable sex differences in the effects of acute stress were recapitulated, along with a substantial genotype difference, in unstressed heterozygous BDNF Val66Met (BDNF$^{Met/+}$) mice, a model of genetic susceptibility to stress[20], which displayed a pre-stressed translational phenotype, i.e., genes, except immediate early genes, were altered in the absence of acute stress. For example, unstressed BDNF Val66Met mice and stressed wild-type mice showed similar alterations in both the glutamatergic and glucocorticoid pathways. In relating these dramatic differences in one neuron type, CA3 neurons, to the overall behaviour of the animal, it is clear that behaviours which require the whole brain cannot be explained by a single cell phenotype. The dramatic differences at the single cell level indicate that the brain is constructed in a way that specific cell types and pathways may be programmed so as to complement and counterbalance each other.

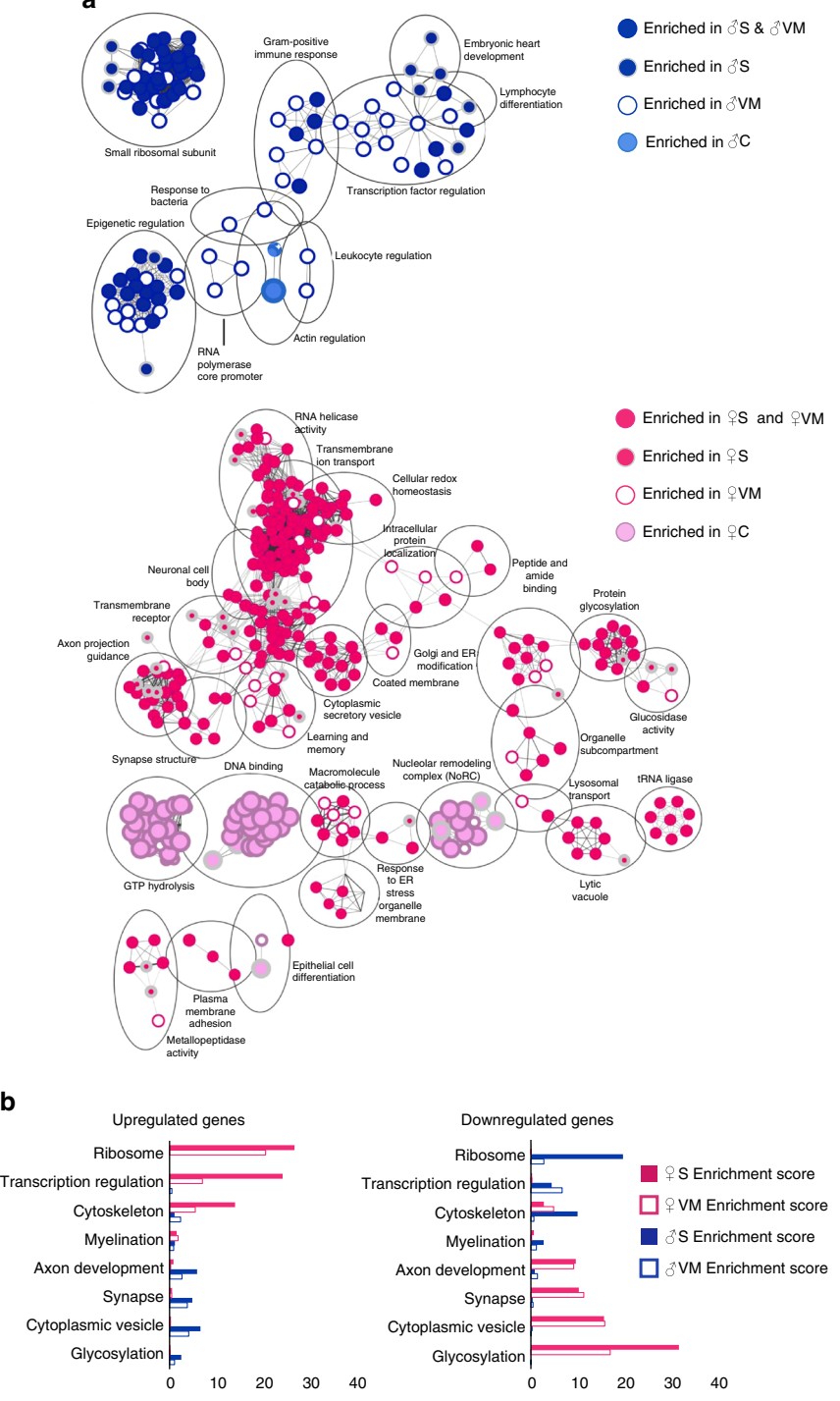

**Fig. 4** The similarity between pathways induced in stressed BDNF$^{+/+}$ mice and those induced in unstressed BDNF$^{Met/+}$ mice are greater in females than in males. **a** Gene set enrichment map of the top 10% most enriched pathways in stressed BDNF$^{+/+}$ males or females (dark blue circles with grey outline and dark pink circles with grey outline, respectively), enriched in unstressed BDNF$^{Met/+}$ males or females (white circle with dark blue outline and white circle with dark pink outline, respectively), enriched in both stressed BDNF$^{+/+}$ and unstressed BDNF$^{Met/+}$ males or females (dark blue circle with dark blue outline and dark pink circle with dark blue outline, respectively), or enriched in unstressed BDNF$^{+/+}$ males or females (light blue circle with light blue outline and light pink circle with light pink outline, respectively) (Z-score < 0.001; absolute fold change > 1.5). Circle size correlates to the number of genes in the pathway. Grey lines connect pathways that share genes. Pathways are clustered based on number of shared genes and named based on biological function. **b** DAVID enrichment scores. Up-regulated or down-regulated genes across the groups in females (pink) and males (blue) are clustered based on their GO scores and given an enrichment score. Eight GO terms showing the highest enrichment score are presented as a bar graph. C unstressed BDNF$^{+/+}$, S stressed BDNF$^{+/+}$, VM unstressed BDNF$^{Met/+}$

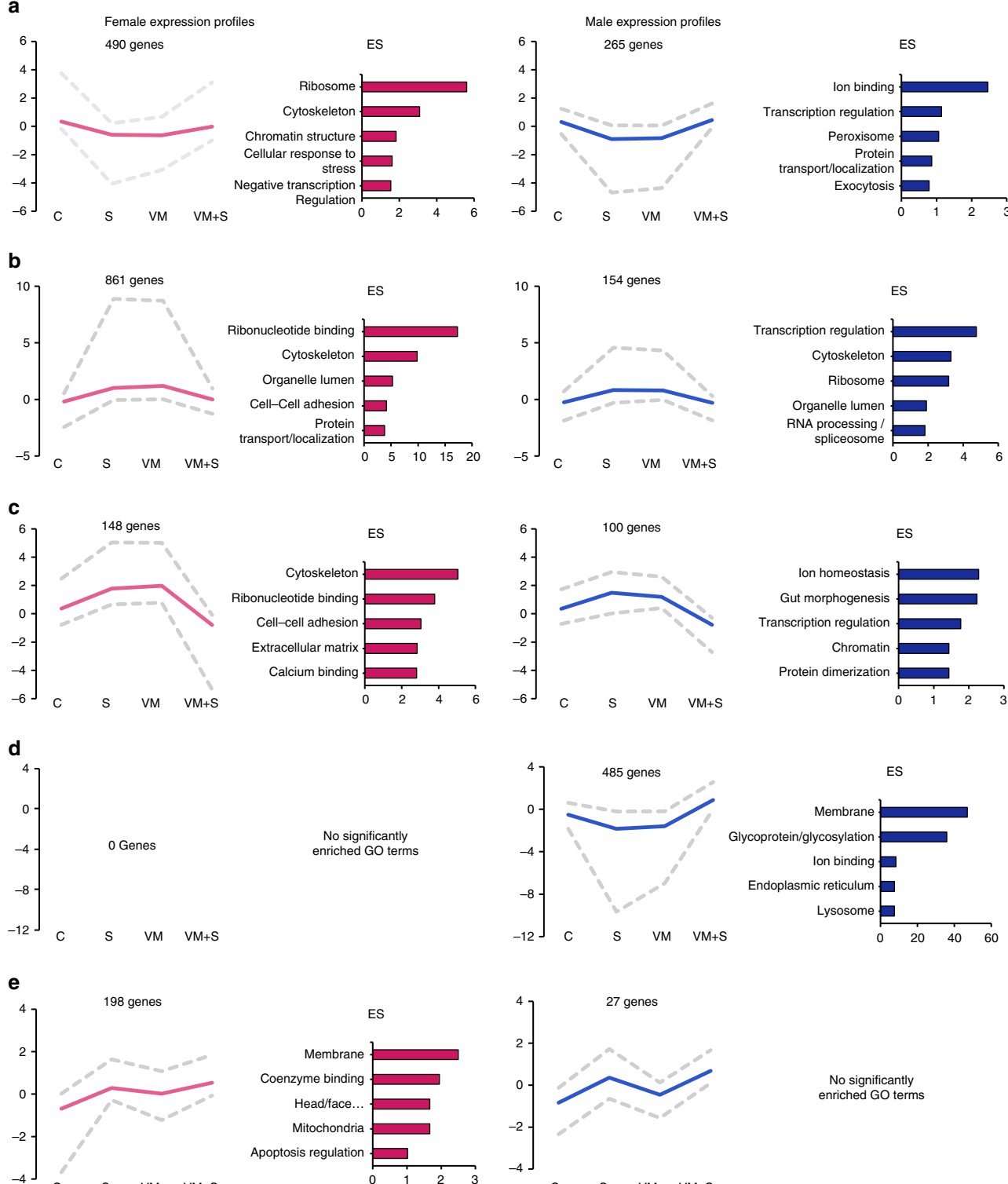

**Fig. 5** Clustering genes by expression profile reveals similar patterns in stressed BDNF$^{+/+}$ and unstressed BDNF$^{Met/+}$. **a–e** Genes were clustered based on their fold change in stressed BDNF$^{+/+}$ (S), unstressed BDNF$^{Met/+}$ (VM), and stressed BDNF$^{Met/+}$ (VM + S) males and females versus unstressed BDNF$^{+/+}$. Fold change direction and significance was determined as follows: up-regulated (z-score ≤ 0.001, fold change ≥ 1.5), down-regulated (z-score ≤ 0.001, fold change ≤ -1.5), or unaltered (z-score ≥ 0.001 and/or absolute fold change ≤ 1.5). Genes with the same expression profiles were clustered together. The maximum, mean, and minimum normalized read density for each clustered profile is presented as a line graph. GO analysis was performed using the DAVID database. The top five most enriched terms for each cluster are presented as a bar graph of their enrichment score (ES). Left: Females, Right: Males

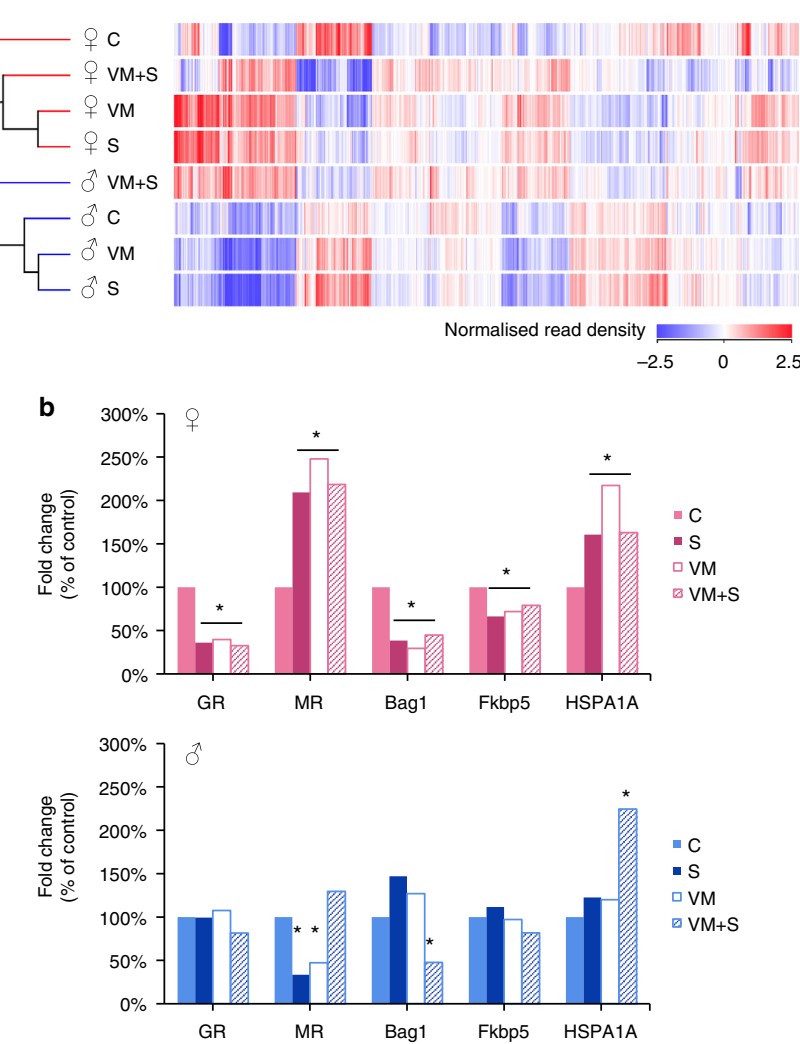

**Fig. 6** Similarities in unstressed BDNF^Met/+ and stressed BDNF^+/+ mice in regard to glucocorticoid receptor-binding genes are sex-specific. **a** Heat map representing the normalized read density of GR-binding genes across all groups in both males and females. The cladogram to the left indicates the similarity in gene expression profiles. **b** Bar charts depicting fold change of selected genes as per cent of unstressed wild type mice in females (left) and males (right). C unstressed BDNF^+/+, S stressed BDNF^+/+, VM unstressed BDNF^Met/+, VM + S stressed BDNF^Met/+

Previous studies have used RNA-seq and microarray in heterogeneous brain tissues or have investigated cell-specific targets using in-situ hybridisation. Here, we utilized the TRAP approach to combine this cell-type specificity with high throughput quantitative genome-wide analysis to investigate the cell-type-specific molecular underpinnings of sex differences in CA3 pyramidal cells. In mouse CA3 pyramidal neurons, we found that acute stress affected the expression of a greater number of genes and pathways in proestrus females than in males, suggesting that stress vulnerability of females may be due to higher transcriptional sensitivity to stress in the hippocampus. This is consistent with previous findings showing that acute stress induced stronger hippocampal gene transcription in females than in males[46–48]. For example, we found that stress in females, but not males, increased expression of genes implicated in the glutamatergic/GABAergic neurotransmission and glycosylation pathways. This supports the observation that developmentally programmed sex differences, and possibly sex hormones, drive gene expression changes in discrete brain regions[49].

We found that sex differences surpassed differences between oestrous cycle stages, and that the oestrus cycle had minimal effect on the translational profile. This could be explained by the lower oestrogen receptor immunoreactivity in the CA3 region compared with other subregions of the hippocampus[50], and also by the largely non-genomic actions of ovarian hormones in the hippocampus[51]. However, in other brain regions, the effect of the oestrous cycle can surpass sex differences. For example, Duclot and Kabbaj[52] have shown that the oestrous cycle surpasses sex differences in regulating the transcriptome of the rat medial prefrontal cortex. While hormonal replacement would elucidate whether oestrous cycle surpasses sex differences, or vice-versa, such an approach would entail stressful survival surgery and the stress of hormone treatment. Stressful experiences alter the translational profile from the naive state, making it difficult to compare to naive intact animals[53, 54].

To further explore sex differences in response to stress in a model of genetic susceptibility to stress, we analysed the translational profile of heterozygous BDNF^Met/+ mice. The CA3 pyramidal neuron transcriptome of unstressed BDNF^Met/+ mice displayed high similarities with gene expression changes observed in wild-type mice after acute stress. Commonalities between pathways and genes induced in stressed wild-type mice and those induced in unstressed BDNF^Met/+ mice were greater in females than in males. GESA and GO analyses of commonly altered genes

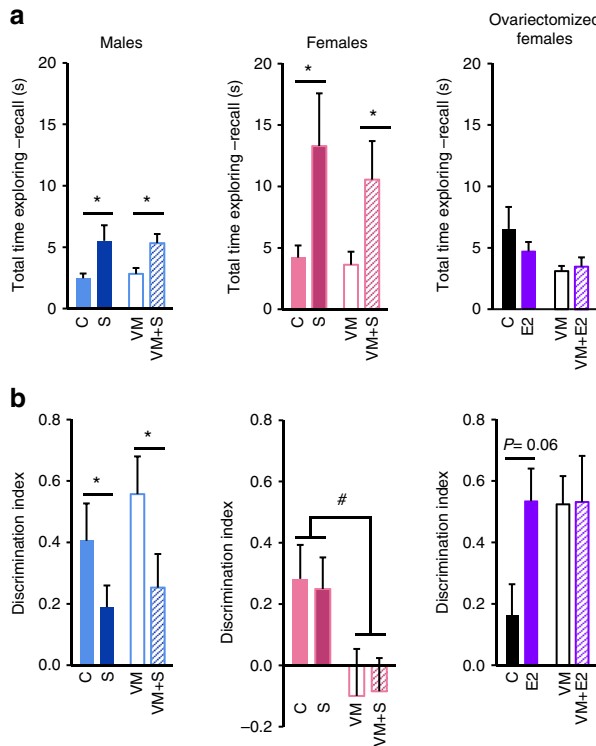

**Fig. 7** The BDNF Met allele interacts with a spatial memory task in females but not in males. **a** Total time exploring either object during the recall phase. **b** Discrimination index ((displaced−stationary)/(displaced + stationary)) of BDNF$^{+/+}$ and BDNF$^{Met/+}$ mice during the recall day. A subset of mice in both groups had been exposed to acute stress before testing. While acute stress affected males and not females, BDNF$^{Met/+}$ females showed a deficit in the discrimination index compared with BDNF$^{+/+}$ females. Values are means + SEM of 8–17 determinations. *$P <$ 0.05 versus the unstressed matched genotype; #$P <$ 0.05 versus wild type. Intact mice: C unstressed BDNF$^{+/+}$ ($n =$ 11 male, $n =$ 17 female), S stressed BDNF$^{+/+}$ ($n =$ 8 male, $n =$ 11 female), VM unstressed BDNF$^{Met/+}$($n =$ 15 male, $n =$ 11 female), VM + S stressed BDNF$^{Met/+}$ ($n =$ 14 male, $n =$ 9 female). Ovariectomized mice: C unstressed ovariectomized BDNF$^{+/+}$ females ($n =$ 8), E2 unstressed ovariectomized BDNF$^{+/+}$ females treated with oestradiol (E2) ($n =$ 9), VM unstressed ovariectomized BDNF$^{Met/+}$ females ($n =$ 10), VM + E2 unstressed ovariectomized BDNF$^{Met/+}$ females treated with E2 ($n =$ 9)

revealed unique pathways in males and females, with females showing alteration of glutamatergic and GABAergic systems. This suggests that that the pre-stress phenotype of heterozygous BDNF $^{Met/+}$ mice is significantly associated with alteration of the inhibitory/excitatory balance. The interplay of BDNF and the glutamatergic system suggests that the BDNF Val66Met mouse model could bridge the gap between the neurotrophic[55] and glutamatergic[56] hypotheses of depression. Glutamatergic neurotransmission is finely tuned by the action of corticosteroids on MR and GR, respectively[57]. It has been shown that corticosterone surge induced by acute footshock stress increases the size of the readily releasable pool of glutamatergic vesicles in the prefrontal cortex, through non-genomic mechanisms involving both MR and GR[58, 59].

Because GR plays a critical role in stress reactivity, we examined how the BDNF Met variant affected a repertoire of GR-binding genes within CA3 neurons of male and female mice. We found that unstressed BDNF$^{Met/+}$ mice displayed a similar transcriptional profile of GR binding genes to that of stressed wild-type mice. However, stress and the BDNF Met variant caused

significantly greater alterations to several genes in females compared with males, such as *Nr3c1* (GR), *Nr3c2* (MR), *Bag1*, *Fkbp5* and *Hspa1a*. Thus we speculate that the pre-stressed transcriptional profile of CA3 neurons is associated with the regulation of GR binding genes, which in turn affect the GABAergic/glutamatergic balance. Further investigation is warranted to examine whether the dramatic sex differences in gene expression we have observed are associated with comparable genome-wide sex differences in protein levels.

The pre-stress profile after acute stress is in line with recent findings by Gray et al.,[29] in which the BDNF$^{Met/+}$ male mice have a translational profile similar to BDNF$^{+/+}$ mice exposed to chronic restraint stress. When female mice were included to investigate gene expression change in CA3 pyramidal neurons, we found that (i) acute stressed induced more genes in females than in males; (ii) the majority of genes that were up-regulated in females were down-regulated in males, and vice-versa; (iii) the pre-stressed phenotype of BDNF$^{Met/+}$ mice was greater in females than in males.

Given the differences in gene expression in CA3 pyramidal neurons, we assessed whether sex differences or the Met allele alter behaviour. Performance in the object placement task relies on short-term and long-term hippocampal spatial memory[44], particularly the CA3 region[40]. Carriers of the BDNF Met allele did not directly recapitulate the behavioural effects of acute stress. In females, the BDNF Met allele interacted with spatial memory performance, with BDNF$^{Met/+}$ females showing an impaired discrimination index regardless of stress. In males, the BDNF Met variant did not affect spatial memory performance. However, acute stress caused an impairment of discrimination index regardless of genotype. Thus, BDNF$^{Met/+}$ females exhibit impaired memory, even in the absence of stress.

We demonstrated that this phenotype was causally associated with circulating ovarian hormones and not recapitulated by oestradiol replacement. Indeed, ovariectomized BDNF$^{Met/+}$ females did not display impairment in the object placement test, resembling unstressed BDNF$^{Met/+}$ males, and oestradiol replacement could not induce the memory impairment in ovariectomized BDNF$^{Met/+}$ females as observed in intact BDNF$^{Met/+}$ females. This suggests that oestradiol alone is not responsible for the impairment in cognitive performance of Met carriers. Rather it is the interplay with other ovarian hormones, the endogenous natural fluctuation of intact mice, and the gene-expression altering effects of the stress of surgery and hormone treatment that induce a unique behavioural phenotype.

Importantly, differences in the translational profile of a single cell type cannot by themselves account for all the sex and genotype behavioural differences. This is not unexpected and suggests that there exists a wide range of sex and genotype differences throughout the brain that result in males and females functioning in similar ways via somewhat different gene expression mechanisms[6–8]. Baseline differences of sex-biased genes have a crucial role in brain development and neurodevelopmental disorders[60], and are widespread in all major brain regions in the adult human brain[61]. Along with the sex-dimorphic gene expression discussed here, there are also functional differences in how brain regions of males and females respond to common experiences[8]. This opens the question of how sex-specific genetic background interacts with environmental stimuli, such as stress, to induce distinct coping strategies in males and females. In humans, women are more likely to develop neuropsychiatric disorders classically associated with stressful experiences[62–64]. One study reported that the sex-specific transcriptional response to stress in the nucleus accumbens contributes to sex differences in stress vulnerability[65]. The authors demonstrated that sub-chronic variable stress in mice alters transcriptional pathways in a

sex-dependent manner, which is associated with behavioural susceptibility in females and resilience in males.

Previous results have shown that the BDNF Met allele interacts with the oestrus cycle in the control of hippocampal function[27], and anxiety-like behaviour[28]. Increases in circulating oestradiol are known to induce hippocampal BDNF mRNA and protein in female mice and rats; and oestradiol requires BDNF signalling to enhance hippocampal synaptic plasticity and dendritic spine formation[66–69]. Because of the importance of BDNF in oestradiol-mediated plasticity, we believe that gonadal hormones modulate the sex-dimorphic pre-stress transcriptional profile of BDNF$^{Met/+}$ mice, inducing higher genetic susceptibility in females. Thus, hormone-regulated gene expression changes in CA3 neurons may reveal novel underpinnings for understanding sex differences in the brain. Investigating the translational profiles of hippocampal subfields, as well as subfields of other brain regions, will pave the way for a better understanding of how brain circuits are programmed as a function of sex and genotype.

## Methods

**Animals**. Male and female C57/BL6 mice aged 3–4 months expressing an Enhanced Green Fluorescent Protein (EGFP) sequence fused to L10a of the large ribosomal subunit under control of a Gprin3 promoter (EGFPL10a) were generated using bacterial artificial chromosomes (BACs), as previously described[29]. BDNFVal66Met knock-in mice were generated in the Lee lab, as previously described[70]. Animals were randomly assigned to stressed or non-stressed groups. One group of animals was unstressed and homozygous for the wild type BDNF allele (BDNF$^{+/+}$; C). The second group of mice were heterozygous for the BDNFVal66Met knock-in allele (BDNF$^{Met/+}$; VM). A third group included C and VM mice that were exposed to acute stress using the forced-swim stressor paradigm (S and VM + S, respectively). Animals were group housed (n = 4–5) in standard cages (28.5 × 17 × 13 cm) and were kept on a 12-h light-dark cycle (lights off 1900hours) in a temperature-controlled room maintained at 21 ± 2 °C. Food and water were available ad libitum. Vaginal cells were collected to identify oestrus cycle stage by flushing the vagina with 70 ul of 1× PBS. Cells were placed on a microscope slide to dry, fixed and stained with Hema3® solution (Fischer, USA), and analysed on a light microscope at ×4 and ×10 magnification. Female mice used for TRAP protocol were grouped according to their oestrus cycle stage: proestrus (high oestradiol), metoestrus/dioestrus (low oestradiol). All procedures were performed in accordance with the National Guidelines on the Care and Use of Animals and a protocol approved by The Rockefeller University Animal Care and Use Committee.

**Surgery and oestradiol replacement**. Two month-old BDNF$^{+/+}$ and BDNF$^{Met/+}$ mice were anaesthetized with a solution of ketamine (75 mg kg$^{-1}$) and xylazine (7.5 mg kg$^{-1}$), hair was removed from each flank using an electric shaver, and the ovaries were removed via a small incision made on each flank. Mice were allowed to recover for 2 weeks before oestradiol replacement began. BDNF$^{+/+}$ and BDNF$^{Met/+}$ mice were treated with either 300 nM β-estradiol in 0.1% ethanol, or with 0.1% ethanol vehicle solution, administered ad libitum in drinking water. Solutions were changed twice weekly, and treatment continued for 6 weeks. Mice were weighed weekly starting from the first day of treatment. Novel object location was assessed during the fourth week of treatment.

**Stress procedure (Forced-swim stress)**. Mice were subjected to a six-minute swimming task in a 4L-graduated cylinder containing 2500 ml of room-temperature water. Control mice were left undisturbed in their home cage. After a 40-minute recovery period, mice were cervical dislocated, immediately decapitated, brains removed and hippocampi were fresh dissected and prepared for TRAP protocol.

**TRAP protocol**. Translating Ribosome Affinity Purification (TRAP) was performed, as described by Heiman et al.[33] Briefly, hippocampal tissue from Gprin3–EFGPL10a mice (n = 5–6 mice per group were pooled for each TRAP to achieve the necessary minimum yield for RNA sequencing) was dissected out and placed in ice-cold dissecting buffer (HBSS, 2.5 mM HEPES- KOH, 35 mM Glucose, 4 mM NaHCO$_3$, 100 µg/ml cycloheximide). Tissue was manually homogenized in homogenisation buffer (10 mM HEPES-KOH, 150 mM KCl, 5 mM MgCl2, 0.5 mM DTT, protease inhibitors, RNasin and Superasin RNAse inhibitors, 100ug/mL cycloheximide) and centrifuged at 2000×g for 10 min at 4 °C, and supernatants were incubated with Nonidet P-40 for 5 min on ice before a 15-min spin at 20,000×g at 4 °C. Supernatant was applied to Streptavidin MyOne T1 Dynabeads (Invitrogen, USA; catalogue no. 65601) that had been incubated with biotinylated protein-L (Pierce, USA; catalogue no. 29997) and α-EGFP antibodies (19C8 and 19F7 from the MSKCC antibody core facility) overnight at 4 °C in 0.15 M KCl buffer containing (10 mM HEPES, 0.15 M KCl, 5 mM MgCl, 1% NP40, 0.05 mM DTT, RNasin RNAase inhibitor, 100 µg/mL cycloheximide). After

immunoprecipitation, unbound fractions were saved, and beads were washed five times with 0.35 M KCl wash buffer (as above with 0.35 M KCl) before beads were resuspended in lysis buffer (Stratagene Absolutely RNA Nanoprep Kit no. 400753). Unbound fraction was isolated using Qiagen Lipid RNA isolation kit (Qiagen, Germany; cat#74804) using a Qiacube (cat#9001292) per the manufacturer's instructions. Quantification and RNA integrity were determined by using a Bioanalyser (Agilent Technologies, USA), and only samples with RNA Integrity Numbers greater than eight (out of 10) were used for sequencing. Specificity of the TRAP protocol was verified by measuring Gprin3 and Gfap expression in RNA sequencing of TRAP fraction versus the unbound fraction (Supplementary Fig. 4).

**RNA-sequencing**. A total of 200 ng of RNA per group was prepared for sequencing by The Rockefeller University Genomics Core Facility using the TrueSeq RNA Sample Preparation Kit v2 (Illumina, USA). A single sequencing library was used per experimental group, comprising of RNA pooled from 5 to 6 animals. The libraries were barcoded to allow for multiplexing within a flow cell lane. Barcoded complementary DNA (cDNA) libraries were sequenced on an Illumina HiSeq 2500 in a single lane to obtain 100-bp single-end reads at an approximate sequencing depth of 35 million reads per sample. The reliability of pooled TRAP-Seq libraries was validated by qRT-PCR in a previous publication[29].

**Object placement task**. A separate set of mice was habituated to an arena (l = 45 cm, w = 45 cm) for two days for 15 min each day. The total distance covered, and the total time spent in the centre were measured using EthoVision® (Noldus, The Netherlands). 24 h later two identical objects (plastic princess dolls) were placed in the arena and the mice were allowed to explore for 10 min. After 24 h one of the objects was moved to the opposing corner of the arena, and mice were allowed to explore the arena for 5 min. The test was videotaped by a camera placed above the arena. The amount of time spent exploring each object was measured using JWatcher™ by an experimenter who was blind to all of the experimental conditions. The discrimination index was calculated as (displaced–stationary)/(displaced + stationary). Animals that showed > 70% preference for one of the objects during acquisition, as well as animals that did not explore the objects during either phase, were excluded from the analysis.

**Sequencing analysis and statistics**. Raw reads were trimmed to remove sequencing artefacts (10 bp from 5′ end) and filtered to remove low quality reads (read with a quality score < 20 in > 10% of bases were discarded) before alignment to mouse genome (mm10) using TopHat2. Filtering reduced the likelihood of false discovery due to errant mapping or trace contaminant in a single sample. Differential expression analysis was conducted with Strand NGS software (Agilent Technologies, USA), in which DESeq was used to quantify transcript reads and obtain Z scores and fold change values for individual genes. Genes with P < 0.001, Benjamini–Hochberg false discovery rate corrected, and fold change greater than 1.5 were selected for further analysis. Differences in integrated read density were visualized against the mouse genome by using the Venn diagram and heatmap tools of Strand NGS. Individual gene expression was visualized using IGV (Broad Institute, USA). GO categories were manually curated from results of the Database for Annotation, Visualization and Integrated Discovery (DAVID) functional annotation cluster tool. Gene cluster enrichment was calculated using Gene Set Enrichment Analysis (GSEA; Broad Institute, USA). The top 10% of the most enriched pathways were plotted as a cluster map using open source software (Cytoscape®). Microsoft Excel (Microsoft, USA) was used to obtain gene expression profiles by sorting genes based on fold change. Behavioural data were analysed using GraphPad Prism (GraphPad Software, Inc., USA) by performing a t-test (two-sided) or a two-way ANOVA followed by Neumann–Keuls post-hoc analysis. A P-value ≤ 0.05 was considered as statistically significant.

**Estimates of effect size and statistical power**. On the basis of previous behavioural studies we have found that a sample size of 8–12 allows us to reliably detect changes of the magnitude we are examining (α = 0.05). Variance is similar among the groups that are being compared.

**Data availability**. The data discussed in this publication have been deposited in NCBI's Gene Expression Omnibus and are accessible through GEO Series accession number GSE100579 (https://www.ncbi.nlm.nih.gov/geo/query/acc.cgi?acc=GSE100579). All other relevant data are available from the authors upon reasonable request.

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

## Acknowledgements

This work was supported by the NIH grant MH102065 to J.D.G., NIH grant MH41256 to B.S.M., NIH grant NS052819 to F.S.L., and the Hope for Depression Research Foundation.

## Author contributions

J.M. and B.S.M. designed the experiments. J.M., G.H.P., M.B.R. and J.F.K. performed the TRAP protocol. J.M. and G.H.P. conducted computational/statistical analysis and behavioural experiments. E.F.S. designed the *Gprin3*-EGFPL10a mouse line. J.D.G., E.M.W. and F.S.L. contributed data analysis. J.M., G.H.P., J.D.G. and B.S.M. wrote the manuscript.

## Additional information

**Competing interests:** The authors declare no competing financial interests.

