## [Peer Review File · Nature Communications]

Reviewers' expertise:

Reviewer #1: Sex difference in the brain, neuroendocrinology, hippocampal-dependent behaviors;

Reviewer #2: Structure and functions of hippocampus, behaviors, BDNF;

Reviewer #3: Gene expression analysis, stress.

Reviewers' comments:

Reviewer #1 (Remarks to the Author):

This is an interesting report from an eminent group in the area of stress physiology and hippocampal function. The authors exploit the power of TRAP technology to characterize gene expression in the CA3 region of the hippocampus with and without acute stress in males versus females and with or without the met allelic variant of BDNF. The significance of the work is the startlingly different gene expression profiles between males and females with and without stress. Although the stress axis is activated in both sexes, and differs only in terms of the height and duration of the corticosterone peak (not shown here but previously demonstrated), there is a remarkable divergence in the short-term gene expression profile. This suggests there are underlying differences in the biology which have remained obscure until now, and yet the phenotype is only modestly shifted, how does this occur? There is a divergence in the effects of stress on spatial learning, as well as an effect of the BDNFmet allele, but that divergence does not appear to be explained by the current data set on gene expression. Thus the immediate functional significance and impact of the findings remains unknown. Nonetheless, the dramatic difference in gene expression profile between males and females is of interest in and of itself and may be one of many future reports to come as scientists in the US increasingly incorporate sex as a biological variable. In this reviewer's opinion the impact of the findings would be better served by focusing on the frank sex differences in gene expression as compared to the modest sex differences in HPA axis activation, instead of the in depth and at times confusing discussion of what gene sets are modified by GR or BDNFmet etc., which obscure the main message. A few other suggestions for improvement are offered below.

1) The number of animals / group used for the TRAP procedure is not clear, were animals pooled or individual? How many?

2) Line 34 – 1st line of the Abstract – what do the authors mean by stating males and females use “distinct brain mechanisms to cope with stress”? As far as I know stress is processed and responded to by all the same brain regions in males and females, there is nothing particularly distinct to either sex, rather it's more a matter of how strong or how long a stress response is manifest. If the authors have something more specific in mind that should be articulated clearly.

3) Line 38 Abstract – what is meant by the “translational effects of acute stress”? Now a days translational usually refers to preclinical findings that may have clinical relevance, not sure how that fits here.

4) Line 44 Abstract – I think this over states the significance of the current results. To be sure the authors have identified an interesting sex difference in gene expression following acute stress and its modulation by the BDNFmet allele, but it does not speak to how the transcriptome is “programmed” or to the “different strategies men and women use to manage stressful experiences”.

5) Line 73 – the rationale for focusing on CA3 is because of its role in episodic and spatial memory, but what about its role in the stress axis?

6) Line 86 – does the BDNFval allele not interact with the estrus cycle? Also it seems odd to say that an allele interacts with steroids to modulate brain function, steroids modulate brain function in part by modulating gene expression and that is where the allele comes into play.

7) Line 93 – the dendritic modeling is more in line with a sex difference than a sex dimorphism. Throughout the manuscript the term “sexual dimorphism” is over used and should be replaced with sex difference, with the possible exception of the remarkable differences in gene expression profile which do seem to be truly dimorphic. The power of that effect will be more evident if the wording is more precise.

8) Line 147 – what is Vulnerabilityco ?

9) Line 147 – what is the difference between BAC-TRAP technology and TRAP technology? If none please be consistent in usage

10) Line 293 – doesn't this result undermine the notion that sex differences in gene expression after stress are leading to sex differences in learning that intersect with the BDNFmet allele. This is addressed briefly in the Discussion but mostly by suggesting further work needs to be done instead of suggesting alternative interpretations.

11) Line 319 – how did you assure cell type specificity? Presumably the cell type is the pyramidal neuron but how were other cells, including glia, microglia and interneurons, excluded? Is this a by-product of the TRAP approach? If so this should be more explicitly stated for those non-familiar with this approach.

12) Figure 1 – its very difficult to see the black numbers in the blue circles

13) Figure 3b – the numbers in the intersecting areas of the Venn diagram are obscured, why are the circles empty here but filled in Figure 1?

14) Figure 5 – the utility of the line graphs in conveying a useful message seems low, not sure what precisely this is supposed to be showing.

15) Line 64 – “both males and females” is redundant

Reviewer #2 (Remarks to the Author):

The manuscript titled "A sexually dimorphic “pre-stressed” translational signature in CA3 pyramidal 2 neurons of BDNF Val66Met mice” by Marocco et al performs RNA-seq on translating RNAs within CA3 hippocampal neurons of both wild type and BDNF Val66Met male and female mice following acute stress. These results provide evidence of drastically differing gene expression patterns in female versus male wild type mice following acute stress. Further, the results reconfirm a baseline “stress-like” phenotype in BDNFVal66Met heterozygotes, and also show that this is more stark in female versus male mice, that have previously been reported to show greater susceptibility to stressors. These results provide interesting evidence for gene-dimorphic translational profiles following acute stress, and also highlight a genotype x stress interaction in the regulation of translating RNAs by acute stress. A previous study from the same group showed distinct translational profiles following acute and chronic stress in CA3 neurons of wildtype and BDNF Val66Met mice. The present study now extends this observation to female mice, highlighting the major differences in male versus female mice.

However, it is unclear whether these changes in gene expression may be associated either via correlation or in a causative manner with changes in cellular function/morphology in CA3 neurons or in behavioral changes following acute stress in both wildtype and BDNF Val66 Met male and female mice.

Specific comments:

1. Given one of the major thrusts of this study is to detail the sexual dimorphism in the translational profiles following acute stress, it would be of value to address the importance of ovarian steroids to mediating these differences. Ovariectomy followed by estrogen replacement could address whether in OVX mice the translational pattern in CA3 neurons resembles males undergoing acute stress. Given prior work (Gray et al., 2016) has already reported an acute stress-evoked CA3 neuron translational profile and also shown a "pre-stressed like" profile in BDNFVal66Met male mice, the major novelty in this study comes from the gender dimorphic pattern reported and it would be necessary to strengthen this observation with additional studies detailing the role of ovarian steroids to the major sexual dimorphism in CA3 translational profiles after acute stress.

2. The authors state that "This indicates that GR binding genes are more sensitive to both stress and the presence of the BDNF Met allele in females than in males". The evidence for the above would be bolstered by ChIP-qPCR experiments to assess whether there is indeed enhanced GR recruitment at specific genes following acute stress in females versus males and with the BDNF Val66Met female and male mice.

3. The behavioral experiments don't provide further insight, rather they appear an afterthought to correlate the extensive RNAseq work performed to behavior. In the absence of an ability to link the observations of altered translational profiles to either changes in cellular architecture/ function or in behavior, it becomes paramount to strengthen the novel observations of sexual dimorphism and provide a deeper understanding for a role for circulating ovarian steroid in driving the far greater extent of gene dysregulation in male versus female mice.

Reviewer #3 (Remarks to the Author):

Marrocco et al provide a genome-wide study of mRNA translation in area CA3 in BDNF Val66Met mice, a mouse model of stress susceptibility. They found that acute stress in WT mice produces a similar translational profile as a SNP that changes valine to methionine at position 66 of the BDNF gene, leading them to conclude that these mice exhibit a "pre-stressed" phenotype. The authors also report extensive sex differences, with a larger number of genes altered by stress in females compared to males. The genes that were changed in both sexes were mostly changed in the opposite direction in males vs females. Females also had greater overlap between differentially-translated genes in response to stress and in the unstressed BDNF Val66Met mice, although only BDNF Val66Met females exhibited memory differences, irrespective of stress.

1. Despite tremendous overlap in translation profiles between unstressed BDNF Val66Met females and stressed WT females in Figure 3, only BDNF Val66Met females exhibited memory deficits (Fig 7), whereas acutely stressed females did not. In contrast, males had fewer differences in translation and less overlap with BDNF Val66Met mice, but both genotypes exhibited similar stress-induced deficits. How do the authors interpret dissimilar behaviours given the extensive similarities in translation in females?

2. The authors state that oestrous phase was controlled for sequencing. Was it controlled for behavioural studies? Variability is much bigger in females than in males.

3. The sex differences reported for stress-induced translation are quite dramatic. Have such dramatic

sex differences been reported for protein levels in these mice, or are differences in translation due to differences in regulation of negative feedback?

4. What was the FDR cutoff?

5. A very small number of genes changed in the same direction with acute stress in males and females, but these genes are not specifically mentioned or analyzed in the text. Since these genes represent the main common factor in response to stress, more information would be beneficial. Which genes overlap? Do they have known function in stress?

6. Excel files containing the genome-wide results of differential comparisons are not included.

Reviewer #1 (Remarks to the Author):

This is an interesting report from an eminent group in the area of stress physiology and hippocampal function. The authors exploit the power of TRAP technology to characterize gene expression in the CA3 region of the hippocampus with and without acute stress in males versus females and with or without the met allelic variant of BDNF. The significance of the work is the startlingly different gene expression profiles between males and females with and without stress. Although the stress axis is activated in both sexes, and differs only in terms of the height and duration of the corticosterone peak (not shown here but previously demonstrated), there is a remarkable divergence in the short-term gene expression profile.

This suggests there are underlying differences in the biology which have remained obscure until now, and yet the phenotype is only modestly shifted, how does this occur? There is a divergence in the effects of stress on spatial learning, as well as an effect of the BDNF_{met} allele, but that divergence does not appear to be explained by the current data set on gene expression. Thus the immediate functional significance and impact of the findings remains unknown. Nonetheless, the dramatic difference in gene expression profile between males and females is of interest in and of itself and may be one of many future reports to come as scientists in the US increasingly incorporate sex as a biological variable. In this reviewer's opinion the impact of the findings would be better served by focusing on the frank sex differences in gene expression as compared to the modest sex differences in HPA activation, instead of the in depth and at times confusing discussion of what gene sets are modified by GR or BDNF_{met} etc., which obscure the main message. A few other suggestions for improvement are offered below.

1) The number of animals/group used for the TRAP procedure is not clear, were animals pooled or individual? How many?

Animals were pooled and this is now stated in the material and methods section (lines 501). Pooling was necessary in order to obtain enough mRNA for the RNA sequencing.

2) Line 34 – 1st line of the Abstract – what do the authors mean by stating males and females use “distinct brain mechanisms to cope with stress”? As far as I know stress is processed and responded to by all the same brain regions in males and females, there is nothing particularly distinct to either sex, rather its more a matter of how strong or how long a stress response is manifest. If the authors have something more specific in mind that should be articulated clearly.

We understand the Reviewer's concern about this statement and we agree that stress is processed by the same brain regions in males and females. However, there is a growing body of evidence that males and females display sex-specific neuroanatomical patterns in response to stress (see Gruene et al., 2015; Derntl et al., 2010; de Vries and Forger, 2015). Unfortunately, word limitations in the abstract does not allow for a more articulated explanation of these important findings. We have rephrased the sentence and replaced “mechanisms” with “circuits” (line 63).

- Gruene TM, Roberts E, Thomas V, Ronzio A, Shansky RM. (2015) Sex-specific neuroanatomical correlates of fear expression in prefrontal-amygdala circuits. *Biol Psychiatry*. 78:186-93.
- Derntl B, Finkelmeyer A, Eickhoff S, Kellermann T, Falkenberg DI, Schneider F, Habel U. (2010) Multidimensional assessment of empathic abilities: neural correlates and gender differences. *Psychoneuroendocrinology*. 35:67-82.
- de Vries GJ, Forger NG. (2015) Sex differences in the brain: a whole body perspective. *Biol Sex Differ*. 6:15.

3) Line 38 Abstract – what is meant by the “translational effects of acute stress”? Now a days translational usually refers to preclinical findings that may have clinical relevance, not sure how that fits here.

Here “translational” is used to report that the RNA sequencing has been performed on translating mRNAs as per TRAP method. We rephrased the statement to clarify this point.

4) Line 44 Abstract – I think this over states the significance of the current results. To be sure the authors have identified an interesting sex difference in gene expression following acute stress and its modulation by the BDNF^{met} allele, but it does not speak to how the transcriptome is “programmed” or to the “different strategies men and women use to manage stressful experiences”.

We agree with the Reviewer. The conclusion of the abstract has been modified to address this concern.

5) Line 73 – the rationale for focusing on CA3 is because of its role in episodic and spatial memory, but what about its role in the stress axis?

We thank the Reviewer for providing us the opportunity to extend the rationale of the study on the role of CA3 in the stress axis. We have now included two studies (Dunn and Orr, 1984; Woolley et al., 1990) describing the role of CA3 in the HPA axis regulation (lines 71-73; 96-98) besides the description of stress-induced atrophy of CA3 neurons.

- Dunn, J. D. & Orr, S. E. Differential plasma corticosterone responses to hippocampal stimulation. *Exp Brain Res* **54**, 1-6 (1984).
- Woolley, C. S., Gould, E. & McEwen, B. S. Exposure to excess glucocorticoids alters dendritic morphology of adult hippocampal pyramidal neurons. *Brain Res* **531**, 225-231 (1990).

6) Line 86 – does the BDNF^{val} allele not interact with the estrus cycle? Also it seems odd to say that an allele interacts with steroids to modulate brain function, steroids modulate brain function in part by modulating gene expression and that is where the allele comes into play.

We understand the Reviewer’s concern about the term “interact”, which here it is not used in a strict sense, but as a synonym for “intersect”. Indeed, we have used this expression to quote the findings by Spencer et al. (2010) and Bath et al. (2012) that demonstrate how estrus cycle is important to unveil behavioral impairments in homozygous BDNF^{Met/Met} mice. This does not mean that BDNF Val allele does not interact with the estrus cycle, but rather emphasize the role of sex hormones in BDNF Met carriers.

- Spencer JL, Waters EM, Milner TA, Lee FS, McEwen BS. (2010) BDNF variant Val66Met interacts with estrous cycle in the control of hippocampal function. *Proc Natl Acad Sci U S A*. 107:4395-400.
- Bath KG, Chuang J, Spencer-Segal JL, Amso D, Altemus M, McEwen BS, Lee FS. (2012) Variant brain-derived neurotrophic factor (Valine66Methionine) polymorphism contributes to developmental and estrous stage-specific expression of anxiety-like behavior in female mice. *Biol Psychiatry*. 72:499-504.

7) Line 93 – the dendritic modeling is more in line with a sex difference than a sex dimorphism. Throughout the manuscript the term “sexual dimorphism” is over used and should be replaced with sex difference, with the possible exception of the remarkable differences in gene expression profile which do seem to be truly dimorphic. The power of that effect will be more evident if the wording is more precise.

We thank the Reviewer in helping us to improve the clarity of the main message of the paper. We have adjusted the language throughout the manuscript to differentiate sex differences from what is truly sex dimorphic.

8) Line 147 – what is Vulnerabilityco?

The typographical error has been corrected.

9) Line 147 – what is the difference between BAC-TRAP technology and TRAP technology? If none please be consistent in usage

There is no difference. The term is now consistent throughout the manuscript.

10) Line 293 – doesn't this result undermine the notion that sex differences in gene expression after stress are leading to sex differences in learning that intersect with the BDNF^{met} allele. This is addressed briefly in the Discussion but mostly by suggesting further work needs to be done instead of suggesting alternative interpretations.

We thank the Reviewer for giving us the opportunity to further our discussion on interpretations of our behavioral results. We have extensively edited our discussion on behavioral sex differences and have added the following sentences in the Discussion, paragraph 1: “In relating these dramatic differences in one neuron type, CA3 neurons, to the overall behaviour of the animal, it is clear that behaviours, which require the whole brain, are not explained by a single cell phenotype. The dramatic differences at the single cell level indicate that the brain is constructed in a way that specific cell types and pathways may be programmed so as to complement and counterbalance each other.” We discuss this more extensively with reference in the revised Discussion (line 349-353). In addition we added a new Fig. 7 that shows how ovariectomy and estrogen replacement fails to replicate behavioral differences between genotypes seen in intact females and possible reasons for this (see response # 3 to Rev 2).

11) Line 319 – how did you assure cell type specificity? Presumably the cell type is the pyramidal neuron but how were other cells, including glia, microglia and interneurons, excluded? Is this a by-product of the TRAP approach? If so this should be more explicitly stated for those non-familiar with this approach.

We thank the Reviewer for this observation that gives us the opportunity to explain the TRAP method in more detail. In the Results section (lines 107-112), we have added a more detailed description of the TRAP method that allows for the isolation of cell-type specific translating mRNAs (Heiman *et al.*, 2014). See also Fig. S4.

- Heiman M, Kulicke R, Fenster RJ, Greengard P, Heintz N. (2014) Cell type-specific mRNA purification by translating ribosome affinity purification (TRAP). *Nat Protoc.* 9:1282-91.

12) Figure 1 – its very difficult to see the black numbers in the blue circles.

The figure has been improved as suggested. Numbers are now white on blue background.

13) Figure 3b – the numbers in the intersecting areas of the Venn diagram are obscured, why are the circles empty here but filled in Figure 1?

The figure has been improved as suggested. The numbers have been moved outside the overlap circle to allow a clearer reading. Full and empty circles represent two different groups, BDNF^{+/+} mice and BDNF^{Met/+} mice, respectively.

14) Figure 5 – the utility of the line graphs in conveying a useful message seems low, not sure what precisely this is supposed to be showing.

We have included this figure to display the common expression profiles of genes across our experimental groups. This type of analysis is useful because it highlights those genes that are being regulated in a pre-stressed manner. To clarify this we have now included the number of genes in each profile. From this type of analysis it can be seen, for example, that a large portion of genes in females are upregulated in both the stressed BDNF^{+/+} and the unstressed BDNF^{Met/+} groups (Fig. 5 B), whereas, in males, there is a major profile that does not exist in females (Fig. 5D). This is now discussed in the result section (lines 243-246).

15) Line 64 – “both males and females” is redundant
The term “both” has been deleted.

Reviewer #2 (Remarks to the Author):

The manuscript titled "A sexually dimorphic “pre-stressed” translational signature in CA3 pyramidal 2 neurons of BDNF Val66Met mice” by Marocco et al performs RNA-seq on translating RNAs within CA3 hippocampal neurons of both wild type and BDNF Val66Met male and female mice following acute stress. These results provide evidence of drastically differing gene expression patterns in female versus male wild type mice following acute stress. Further, the results reconfirm a baseline “stress-like” phenotype in BDNFVal66Met heterozygotes, and also show that this is more stark in female versus male mice, that have previously been reported to show greater susceptibility to stressors. These results provide interesting evidence for gene-dimorphic translational profiles following acute stress, and also highlight a genotype x stress interaction in the regulation of translating RNAs by acute stress. A previous study from the same group showed distinct translational profiles following acute and chronic stress in CA3 neurons of wildtype and BDNF Val66Met mice. The present study now extends this observation to female mice, highlighting the major differences in male versus female mice. However, it is unclear whether these changes in gene expression may be associated either via correlation or in a causative manner with changes in cellular function/morphology in CA3 neurons or in behavioral changes following acute stress in both wildtype and BDNF Val66Met male and female mice.

Specific comments:

1. Given one of the major thrusts of this study is to detail the sexual dimorphism in the translational profiles following acute stress, it would be of value to address the importance of ovarian steroids to mediating these differences. Ovariectomy followed by estrogen replacement could address whether in OVX mice the translational pattern in CA3 neurons resembles males undergoing acute stress. Given prior work (Gray et al., 2016) has already reported an acute stress-evoked CA3 neuron translational profile and also shown a “pre-stressed like” profile in BDNFVal66Met male mice, the major novelty in his study comes from the gender dimorphic pattern reported and it would be necessary to strengthen this observation with additional studies detailing the role of ovarian steroids to the major sexual dimorphism in CA3 translational profiles after acute stress.

We thank the Reviewer for highlighting that the novelty of the current work lies in the sex-dimorphic gene expression in the pre-stressed state of BDNF Met carriers. Indeed, Gray et al., 2016, which neither include the same comparative gene-expression analysis as presented here, nor discusses acute stress in BDNF^{Met/+} mice, indicated the need for further characterizations of the pre-stressed translational phenotype. We would also like to thank the Reviewer for pointing out the importance of ovarian steroid hormones in sex-dimorphic gene expression.

We have included a new figure that shows that the effects of fluctuating endogenous levels of ovarian hormones do not produce dramatic differences in stress-induced gene expression change in CA3 (Fig. S2). This is now reported in the result section (lines 115 and 134-142). High and low oestradiol females displayed 4,676 genes that were commonly altered by acute stress, comprising 86.8% of genes altered in high oestradiol females and 75.3% of genes altered in low oestradiol females (Fig. S2A). This can perhaps be explained by reduced sensitivity to estradiol of CA3 neurons compared to other hippocampal regions. We note that the CA3 does have non-genomic forms of the classical estrogen receptor, and is thus estrogen-responsive, but not to the same extent of other brain regions (at least in regards to transcription). Non-genomic actions of estradiol and progesterone are primarily responsible for the cyclic turnover of spine synapses in the hippocampus (Li et al., 2004; McEwen and Woolley, 1994).

Moreover, ovariectomy and hormone replacement only crudely recapitulate some of the effects of the natural cyclic variation of ovarian and other hormones and their effects on behavior (see response to question #3: behavioral results). It

is important to notice that stress of surgery associated with gonadectomy and loss of gonads, as well as stress of hormone or vehicle treatment, would create a background of gene expression that is different from that of the present paper, where animals were “naïve” up to the point of acute stress (Gray et al., 2014). Based on our previous work, we would expect a different pattern of gene expression for each sex and treatment group. Although of some interest, this would not enhance the main point of our paper.

Finally, a thorough gonadectomy and hormone replacement study would require, minimally, intact and gonadectomized males and females and replacement with testosterone of the males and estradiol of the females - plus separate groups of each that undergo acute stress. Even then, the females might require groups given estradiol plus progesterone sequentially to mimic the proestrus phase plus or minus acute stress. This experimental protocol would require an entire new set of double transgenic mice along with RNA sequencing runs and a computational analysis that would be several times greater than the one we have presented. Yet, the amount of data would be significant and require a separate paper and, while interesting in its own right, it would not add to our main point cited above.

- Li C, Brake WG, Romeo RD, Dunlop JC, Gordon M, et al. 2004. Estrogen alters hippocampal dendritic spine shape and enhances synaptic protein immunoreactivity and spatial memory in female mice. *Proc. Natl. Acad. Sci. USA* 101: 2185-90
- McEwen BS, Woolley CS. 1994. Estradiol and progesterone regulate neuronal structure and synaptic connectivity in adult as well as developing brain. *Exp. Geront.* 29: 431-36
- Gray JD, Rubin TG, Hunter RG, McEwen BS. 2014 Hippocampal gene expression changes underlying stress sensitization and recovery. *Mol Psychiatry.* 19: 1171-8.

2. The authors state that “This indicates that GR binding genes are more sensitive to both stress and the presence of the BDNF Met allele in females than in males”. The evidence for the above would be bolstered by ChIP-qPCR experiments to assess whether there is indeed enhanced GR recruitment at specific genes following acute stress in females versus males and with the BDNF Val66Met female and male mice.

We thank the Reviewer for drawing our attention to this statement. We agree that, as previously written, a ChIP-qPCR experiment would be required to assess whether there is enhanced GR recruitment after acute stress and in the Met genotype. We have rephrased the statement to more accurately convey our intended message, which is that stress and the Met allele induce a greater change of GR binding gene expression in females than in males (line 293-295).

3. The behavioral experiments don’t provide further insight, rather they appear an afterthought to correlate the extensive RNAseq work performed to behavior. In the absence of an ability to link the observations of altered translational profiles to either changes in cellular architecture/function or in behavior, it becomes paramount to strengthen the novel observations of sexual dimorphism and provide a deeper understanding for a role for circulating ovarian steroid in driving the far greater extent of gene dysregulation in male versus female mice.

We would like to point out that we report data on one neuron type in the brain but behavior requires a more complex neuronal network. As far as sex differences in behavior, studies like that of Derntl et al. (2010) and Shansky et al. (2010) show that males and females do many of the same things equally well but use different pathways in the brain. In that context, we believe that the behavioral experiments that were performed in parallel with our RNA sequencing experiments do provide further insights when looked at from the standpoint of sex differences and effect of the BDNF Met allele. What our CA3 RNA-seq data shows for this one neuron type is an unexpected, dramatic difference in gene type and direction of change with acute stress. Yet the behavior of the animals shows overall effects of sex and genotype. These are not explained by a single cell phenotype, nor would we expect them to be. So the dramatic differences at the single cell level

indicate that the brain is constructed in a way that specific cell types and pathways are programmed in ways that complement and counterbalance each other. This is now emphasized in the Discussion.

As far as the role of ovarian hormones, we note above in Response #1 that estrous cycle differences have a minimal effect on the stress effects on gene expression in CA3 neurons. More broadly, we have pointed out in Response #1 above that understanding ovarian hormone contributions is a more complex issue than the reviewer suggests because of the gene-expression altering effects of the stress of surgery and hormone treatment, along with the fact that estradiol replacement does not replace natural ovarian cycling. To illustrate this, we included new experiments that i) highlight the role of circulating ovarian hormones in the cognitive behavioral impairment selectively observed in heterozygous $BDNF^{Met/+}$ females, and ii) demonstrate the inadequacy of estrogen treatment after ovariectomy to produce the same phenotype as intact mice (see figure below adapted from Fig. 7B). We found that ovariectomized $BDNF^{Met/+}$ mice treated with vehicle (solid white bar, black outline) did not exhibit memory impairment in the novel object placement test, resembling unstressed $BDNF^{Met/+}$ males (solid white bar, blue outline). Estradiol replacement improved memory in $BDNF^{+/+}$ (solid purple bar) but had no effects in $BDNF^{Met/+}$ (solid white bar, purple outline), which still displayed no impairment in cognitive performance (Fig 7B in manuscript). These findings indicate that memory impairment in $BDNF$ Met carriers is associated with circulating ovarian hormones, but estradiol replacement alone, after ovariectomy, did not reestablish the impaired phenotype, suggesting a role for other ovarian hormones as well as possible alterations due to the surgery of ovariectomy and hormone replacement. Indeed, estradiol replacement after ovariectomy could not recapitulate the behavioral phenotype of intact mice with intact ovaries and estrous cycles (lines 321-336). This supports our rationale in our Response #1 above regarding this issue of hormone replacement. Finally, we have extensively edited our discussion section to provide further insight into how transcriptional modifications could alter behavioral response in a sex-by-genotype manner (see in particular 349-353; 422-446).

Reviewer #3 (Remarks to the Author):

Marrocco et al provide a genome-wide study of mRNA translation in area CA3 in BDNF Val66Met mice, a mouse model of stress susceptibility. They found that acute stress in WT mice produces a similar translational profile as a SNP that changes valine to methionine at position 66 of the BDNF gene, leading them to conclude that these mice exhibit a “prestressed” phenotype. The authors also report extensive sex differences, with a larger number of genes altered by stress in females compared to males. The genes that were changed in both sexes were mostly changed in the opposite direction in males vs females. Females also had greater overlap between differentially-translated genes in response to stress and in the unstressed BDNF Val66Met mice, although only BDNF Val66Met females exhibited memory differences, irrespective of stress.

1. Despite tremendous overlap in translation profiles between unstressed BDNF Val66Met females and stressed WT females in Figure 3, only BDNF Val66Met females exhibited memory deficits (Fig 7), whereas acutely stressed females did not. In contrast, males had fewer differences in translation and less overlap with BDNF Val66Met mice, but both genotypes exhibited similar stress-induced deficits. How do the authors interpret dissimilar behaviours given the extensive similarities in translation in females?

We thank the Reviewer for bringing this point to our attention. We have expanded our discussion section to include alternative interpretations of our behavioral results, and how they might reflect our translational data. We believe that the dissimilarity between translational profile and behavior is not surprising when accounting for differences in immediate early gene activation and corticosterone levels, which both require applied stressors regardless of genotype. In our opinion it is indeed expected that animals undergoing an acute stress (and handling) 40 minutes prior to testing will exhibit different behavior from unstressed animals, even if they otherwise have a similar gene expression profile. We call attention to our response #3 to reviewer 2.

2. The authors state that oestrous phase was controlled for sequencing. Was it controlled for behavioural studies? Variability is much bigger in females than in males.

Yes, we did control for the estrous cycle in our behavioral studies, and found no effect of estrous cycle stage on discrimination index. However, we would like to point out that the behavioral paradigm that we have used spans over 4 days, namely 2 days of habituation, 1 day of acquisition, and 1 day of testing. We were able to collect vaginal smear only during the final day, as doing so on days 1-3 would introduce unwanted stressors during the testing paradigm. One would also consider that the effect of estrous cycle could impact behavior during any of the three phases of the behavior test (4 days total), hence requiring a study of the effects of the estrous cycle stages on habituation, acquisition, and recall. One interesting point the Reviewer may notice is that, in this new version of the manuscript, we have included a new figure 7 in which we have tested the role of ovarian hormones in the novel object location performance. Our findings indicate that memory impairment in BDNF Met carriers is possibly associated with circulating ovarian hormones, because ovariectomized BDNF^{Met/+} do not show cognitive deficit. However, estradiol replacement alone could not recapitulate the behavioral phenotype of intact mice that likely depends on endogenous fluctuation of estradiol in concert with other gonadal hormones (lines 321-336). Again we discuss this in response #3 to Rev 2.

3. The sex differences reported for stress-induced translation are quite dramatic. Have such dramatic sex differences been reported for protein levels in these mice, or are differences in translation due to differences in regulation of negative feedback?

We agree with the Reviewer that considering protein levels is an important next step, and we have now added a statement commenting on this (lines 402-404). However, to achieve a protein analysis of similar scale we would need to

conduct a genome-wide proteomic analysis on isolated CA3 pyramidal cells. To our knowledge such analysis has not been performed and is beyond the scope of the present study.

4. What was the FDR cutoff?

The FDR cutoff is 0.001 (line 545-546).

5. A very small number of genes changed in the same direction with acute stress in males and females, but these genes are not specifically mentioned or analyzed in the text. Since these genes represent the main common factor in response to stress, more information would be beneficial. Which genes overlap? Do they have known function in stress?

We thank the Reviewer for this suggestion. We have added a supplementary figure, which details the genes in questions (Fig. S1, lines 122-124). Genes with known function in stress are in bold.

6. Excel files containing the genome-wide results of differential comparisons are not included.

An excel file containing the genome-wide results of differential comparisons is now included as supplementary material.

Reviewers' comments:

Reviewer #1 (Remarks to the Author):

The authors have done a good job of responding to the concerns of the reviewers. This is one of those studies that raises more questions than it answers but that is ok. I'm not sure the addition of the supplemental data on estradiol treatment is a plus given the unusual manner of administration (drinking water) and no independent verification of what circulating levels were achieved, but overall this study is not about hormones effects, its more about sex differences and those two are not the same thing

Reviewer #2 (Remarks to the Author):

The authors have addressed the issues raised and their inclusion of data from different stages of the estrous cycle supports their contention that OVX experiments with simple E2 replacement may not fully allow them to explain their sex-dimorphic effects on acute stress evoked CA3 translational profiles. This data is useful and allows for a richer interpretation. Further the discussion has been strengthened.

Reviewer #3 (Remarks to the Author):

Most of my concerns were adequately addressed. However, the manuscript is still missing the full data set in Excel, even though the authors say it has been submitted. In addition, there is no reference to data being deposited through GEO, nor a reference to the accession number.

Additional comments:

After looking at the file, I have more questions for the authors and I am much less enthusiastic about the manuscript. There is no explanation to go with the Excel file, so I am making some assumptions on the data. There is only one column per experimental group, which leads me to believe that there were no replicates included in the analysis (I.e. that only one sample was sequenced per group). The manuscript doesn't provide any details about how many samples were sequenced per group, but the authors replied to Reviewer 1's first question by saying that they had to pool tissue from several mice to get enough RNA. While pooling tissue is entirely reasonable in this scenario, the pooled RNA still results in only one sequenced sample per group, which is entirely unrealistic for proper statistical analyses. In this paper <https://www.ncbi.nlm.nih.gov/pmc/articles/PMC4728800/>, Conesa et al (2016) provide basic guidelines for bioinformatics and state that a generally, a minimum of 3 replicates per group are required (see Figure 1). With only 1 replicate, I cannot recommend that this paper is published. However, it is possible that I am misreading the excel sheet and that the data show an average across replicates. If this is true, the authors should state so explicitly.

We thank the Reviewers for further considering our revised paper (attached) and our response to their comments. Reviewer 1 and 2 acknowledged our effort to improve the paper and endorsed its publication. However, Reviewer 3 raised some new concerns about the statistical strength of our RNA-Seq data, and we would like to address these concerns and clarify the choice of our experimental design and analytical approach.

Reviewer#1

We thank the Reviewer for the positive comments. We agree that estradiol in drinking water is not a usual manner of administration. We used this method to minimize the stress of injection (which is a more common protocol to replace estradiol in rodents). We follow the protocol by Levin-Allerhand *et al.* (2005) and validated the success of replacement by measuring the uteri weight at the end of the treatment (data not shown). As the reviewer points out, this is not a paper on sex hormones and we will make sure to include this data in our next article on hormonal manipulation.

Reviewer#2

We thank the Reviewer for the positive comments and for appreciating the new results on estradiol replacement.

Reviewer#3

We used RNA-Seq libraries, in which each condition was comprised of a pool of ribosomal immunoprecipitated mRNA from 5-6 animals (manuscript lines 503-504; 527-528). Indeed, generating additional replicate sequencing libraries would improve the statistical power of the analysis. However, doing so in this study would have been prohibitive because of the number of double transgenic animals (6 mice/group=216 mice) required to generate multiple pools for each condition as well as the significantly increased sequencing costs resulting from additional pools. We emphasize in our study that the reliability of pooled TRAP-Seq libraries was validated by qRT-PCR in a previous publication from our lab (Gray *et al.*, 2016) (manuscript line 531). In this published paper, the same TRAP-Seq methodology used here was directly validated against qRT-PCR data for the detection of differential expression of genes in response to acute stress. Previously, n=4 independent TRAP immunoprecipitations with n=2 mice/TRAP was compared against 1 TRAP-Seq library of n=6 mice for 8 genes to show that the fold change from these independent experiments correlated with each other at an $R^2 = 0.79$. This demonstrates that our data has high reliability with other techniques for quantifying mRNA, despite the lack of technical replicates.

Moreover, as stated in our revised manuscript, in order to further improve the reliability of the data, we incorporated a number of modifications to our sequencing design to improve the power of this study over a typical RNA-Seq experiment conducted in heterogeneous tissue samples. *First*, the TRAP-immunoprecipitation and the polyA library prep are two independent steps designed to ensure that we have a homogenous transcriptome (from a single cell type, CA3 excitatory pyramidal neurons) and that we have eliminated rRNA and any small RNAs. This substantially decreases the diversity of the transcriptome and thereby improves power at equivalent sequencing depths (Heiman *et al.*, 2008) (manuscript lines 112-114).

Second, we have increased read length to 100bp and sequenced each sample at a depth of 35-40M reads each (manuscript lines 529-530) to improve transcriptome coverage beyond typical studies using 50bp reads at a depth of 10M reads (Wang *et al.*, 2011), thereby increasing the power of the comparisons. Reducing transcriptome diversity minimizes the number of comparisons and increasing sequencing depth further improves our statistical power. Third, we removed genes with fewer than 20 reads/gene to further reduce the overall number of comparisons, thus reducing the likelihood of a false discovery due to errant mapping or trace contaminants in a single sample (manuscript lines 547-549). We also set a higher significance threshold of $p < 0.001$ with false discovery correction and a fold change cutoff of 1.5, to further minimize false discoveries (manuscript lines 551-552).

As a result of this, it is not surprising that the data presented in our manuscript *picked up and further validated changes in a number of genes already known to be altered by acute stress (cFos, Arc, etc.) that have been established by in-situ hybridization and qRT-PCR in the stress literature for the past two decades.*

Finally, as noted by the reviewers, we would like to emphasize the novelty of our findings, so that their publication will pave the way for further investigation on sex differences in gene expression in specific brain regions. To our knowledge this is the first time an investigation of this nature has been reported.

The data discussed in this publication have been deposited in NCBI's Gene Expression Omnibus (Edgar *et al.*, 2002) and are accessible through GEO Series accession number GSE100579 <https://www.ncbi.nlm.nih.gov/geo/query/acc.cgi?acc=GSE100579>. (manuscript lines 564-567)

References

- Levin-Allerhand JA, Sokol K, Smith JD. 2003. Safe and effective method for chronic 17beta-estradiol administration to mice. *Contemp Top Lab Anim Sci.* 42:33-5.
- Heiman M, Schaefer A, Gong S, Peterson JD, Day M, et al. 2008. A translational profiling approach for the molecular characterization of CNS cell types. *Cell* 135: 738-48
- Wang Y, Ghaffari N, Johnson CD, Braga-Neto UM, Wang H, et al. 2011. Evaluation of the coverage and depth of transcriptome by RNA-Seq in chickens. *BMC Bioinformatics* 12 Suppl 10: S5

- Gray JD, Rubin TG, Kogan JF, Marrocco J, Weidmann J, et al. 2016 Translational profiling of stress-induced neuroplasticity in the CA3 pyramidal neurons of BDNF Val66Met mice. *Mol Psychiatry*. Dec 13.
- Edgar R, Domrachev M, Lash AE. 2002 Gene Expression Omnibus: NCBI gene expression and hybridization array data repository. *Nucleic Acids Res*. 30:207-10.

REVIEWERS' COMMENTS:

Reviewer #3 (Remarks to the Author):

The additional manuscript text and author replies address my concerns.

Reviewer #3

The additional manuscript text and author replies address my concerns.

We are pleased that Reviewer #3 appreciated our efforts to make the data analysis as rigorous as possible.